# Neuromodulation of the cerebellum rescues movement in a mouse model of ataxia

Lauren N. Miterko[1,2,3], Tao Lin [1,3], Joy Zhou[1,3,4], Meike E. van der Heijden [1,3], Jaclyn Beckinghausen[1,3,4], Joshua J. White[1,3,4] & Roy V. Sillitoe [1,2,3,4,5 ✉]

Deep brain stimulation (DBS) relieves motor dysfunction in Parkinson's disease, and other movement disorders. Here, we demonstrate the potential benefits of DBS in a model of ataxia by targeting the cerebellum, a major motor center in the brain. We use the *Car8* mouse model of hereditary ataxia to test the potential of using cerebellar nuclei DBS plus physical activity to restore movement. While low-frequency cerebellar DBS alone improves *Car8* mobility and muscle function, adding skilled exercise to the treatment regimen additionally rescues limb coordination and stepping. Importantly, the gains persist in the absence of further stimulation. Because DBS promotes the most dramatic improvements in mice with early-stage ataxia, we postulated that cerebellar circuit function affects stimulation efficacy. Indeed, genetically eliminating Purkinje cell neurotransmission blocked the ability of DBS to reduce ataxia. These findings may be valuable in devising future DBS strategies.

[1] Department of Pathology and Immunology, Baylor College of Medicine, Houston, TX, USA. [2] Program in Developmental Biology, Baylor College of Medicine, Houston, TX, USA. [3] Jan and Dan Duncan Neurological Research Institute of Texas Children's Hospital, Houston, TX, USA. [4] Department of Neuroscience, Baylor College of Medicine, Houston, TX, USA. [5] Development, Disease Models & Therapeutics Graduate Program, Baylor College of Medicine, Houston, TX, USA. ✉email: sillitoe@bcm.edu

Deep brain stimulation (DBS) is an FDA-approved, neurosurgical technique used to treat motor and non-motor diseases including Parkinson's disease, dystonia, tremor, and obsessive–compulsive disorder. DBS strategies have targeted the thalamus, globus pallidus, subthalamic nucleus, and nucleus accumbens. DBS benefits depend on the parameters used and on the symptoms being treated[1]. For instance, in dystonia, DBS can be ~85% or <20% effective[2]. While technical challenges such as electrode drift impact patient responsiveness[3], other problems such as motor decline[4] and tolerance[5] suggest suboptimal brain targeting. Indeed, stimulating different targets improves medically-refractory dystonia[6]. We hypothesize that stimulating the motor regions that contribute most to key symptoms at the time of treatment could improve behavioral outcomes.

To test this hypothesis, we targeted DBS to the cerebellum, a primary motor control center, and asked whether stimulation restores movement in ataxia. The choice of the cerebellum as the target, and ataxia as the symptom, was motivated by several reasons. First, cerebellar outflow circuits have major therapeutic potential since they connect the cerebellum with more than three-dozen brain and spinal cord nuclei[7]. Critical computational sites in the cerebellar circuit are also experimentally accessible and highly plastic, which, in theory, could support long-term, behavioral improvements[8]. Moreover, the cerebellum plays a pathogenic role in a growing list of ataxias, including spinocerebellar ataxia, Friedreich's ataxia, ataxia telangiectasia, and episodic ataxia—all of which are characterized by a progressive decline in motor function and eventually, immobility. Contrary to the sometimes variable effects of improving basal ganglia function to reduce dystonia, improving cerebellar function universally recovers mobility in different ataxias[9–13], partly because the cerebellum is distinctly vulnerable to hereditary and non-hereditary insults. Illustratively, the principle cell type of the cerebellar circuit, the Purkinje cell, selectively degenerates in most forms of ataxia, as do the downstream cerebellar and brainstem nuclei[14]. Accordingly, a major goal of ataxia research has been to elucidate the molecular mechanisms underlying this specificity, which could then be used to tailor pharmaceuticals and improve the efficacy of gene and stem cell therapies. However, there are many practical challenges to applying these therapies in vivo, including efficient delivery of the agent to a large number of affected neurons[15], correcting complex cellular functions and animal behaviors[16], and maintaining the benefits throughout life[17].

We argue that DBS targeted to the cerebellum bypasses several critical hurdles. While stimulating the cerebellar cortex was inconsistent for treating motor deficits[7], stimulating the cerebellar nuclei has been more efficacious (EDEN; Improvement of Upper Extremity Hemiparesis Due to Ischemic Stroke: A Safety and Feasibility Study, ClinicalTrials.gov Identifier: NCT02835443)[7,18]. Here, we investigated whether DBS combined with physical activity enhances motor function in the *Car8 waddles* (*Car8wdl*) mutant mouse that models a form of hereditary ataxia in humans[19–21]. Our study was motivated by the presence of carryover effects observed after DBS is terminated in patients[22,23]. We therefore asked whether such carryover effects can be controlled in a manner that would provide deliberate, long-term benefits to behavior. Particularly valuable to our study is that cerebellar circuit function is altered in *Car8wdl* mice, but the cells do not develop overt anatomical defects and they do not degenerate[24,25]. Therefore, cerebellar neural circuitry is experimentally accessible and amenable to treatment and post-stimulation tracking. Our approach tests the therapeutic potential of the cerebellum by targeting DBS to the interposed nuclei. We show that cerebellar DBS restores motion in ataxia and that the rescue of motor behavior was the greatest when the treatment is paired with exercise and starts early after the onset of ataxia.

## Results

### Beta-frequency DBS into the cerebellum improves *Car8wdl* motor behavior.

We recently developed a DBS strategy that targets the interposed cerebellar nucleus[26,27], which sends major connections to the red nucleus and thalamus to control ongoing motor function. Unlike the dentate or fastigial nucleus, the interposed nucleus facilitates locomotor adaptation[28] and contains cerebellospinal neurons, which synaptically target pre-motor circuits in the spinal cord[29]. Feedback collaterals from the red nucleus also selectively innervate the interposed nucleus[30], suggesting that the interposed nucleus can independently process sensorimotor information and influence corollary discharge. Since balance, coordination, locomotor adaptation, and sensorimotor processing degrade in ataxia, we reasoned that the interposed nucleus could be an ideal target for restoring movement. The impact of DBS specifically on gait in ataxia, as demonstrated in a recent study[31], additionally motivated our design of a therapeutic approach that might boost its efficacy. This included stimulating the interposed nuclei of ataxic *Car8wdl* mice (Fig. 1a–d; Supplementary Figs. 1–2) while the mice were actively engaged on an accelerating rotarod. In this paradigm, the rotarod assay serves as a proxy for challenging exercise, since it is a demanding physical task that also requires skilled learning. We pair cerebellar stimulation with exercise because previous attempts to restore function in Parkinson's disease and stroke show that combining neuromodulation with motor rehabilitation significantly enhances therapeutic outcomes[32,33]. However, this combinatorial approach has only been tested in severe neurodegenerative diseases, where plastic changes in surviving circuits, such as compensatory miswiring after axonal regression and significant alterations in dendritic architecture, occur. It is unclear if additional therapeutic benefits could be unlocked for ataxia if physical activity is combined with DBS when circuits are dysfunctional, but anatomically intact, or when DBS is targeted to a cerebellar region that contains circuitry that is specifically involved in sensorimotor processing and locomotor adaptation during ongoing, intentional movements.

Our behavioral paradigm consisted of two periods: "Before ±DBS" and "With or Without DBS" (Fig. 1a, b). We measured the latency to fall off of the accelerating rotarod for *Car8wdl* mice during both of these periods to calculate their improvement during stimulation at various frequencies. We specifically stimulated the cerebellar nuclei at 0, 2, 13, and 130 Hz (Fig. 1e, f; Supplementary Fig. 1), given that these frequencies support distinct aspects of normal circuit function. For instance, delta- and theta-frequencies between 1 and 9 Hz promote cerebellar learning at climbing fiber-to-Purkinje cell as well as parallel fiber-to-Purkinje cell synapses[34]. In contrast, beta-frequencies from 10 to 30 Hz facilitate communication in the cerebello-thalamo-cortical pathway, and higher frequencies between 30 and 260 Hz promote scaling, planning, and neural synchronization[34].

We found that *Car8wdl* mice improve on the rotarod with cerebellar DBS, but only when it is delivered at a beta-frequency (Fig. 1e, f; Supplementary Fig. 1c; Supplementary Movie 1; 13 Hz, $65.29 \pm 25.11\%$, $n = 12$, $p = 0.0095$). We conducted several control experiments to substantiate these effects. Without surgery, *Car8wdl* mutant mice do not significantly improve (Fig. 1e, f; Supplementary Fig. 1c; $11.81 \pm 14.08\%$, $n = 8$; $p > 0.9999$). Similarly, electrode implantation does not significantly alter *Car8wdl* motor performance (Fig. 1e, f; Supplementary Fig. 1c; Supplementary Movie 2; 0 Hz, $39.76 \pm 13.66\%$; $n = 12$, $p = 0.8421$), nor does stimulating at 2 or 130 Hz (Fig. 1e, f; Supplementary Fig. 1c; 2 Hz, $49.54 \pm 18.36\%$, $n = 6$, $p = 0.9869$; 130 Hz, $23.52 \pm 17.67\%$, $n = 12$, $p = 0.9796$). Control mice modestly improve from their baseline performances during DBS, with this improvement showing statistical significance at 0, 2, and 130 Hz (Supplementary Fig. 3;

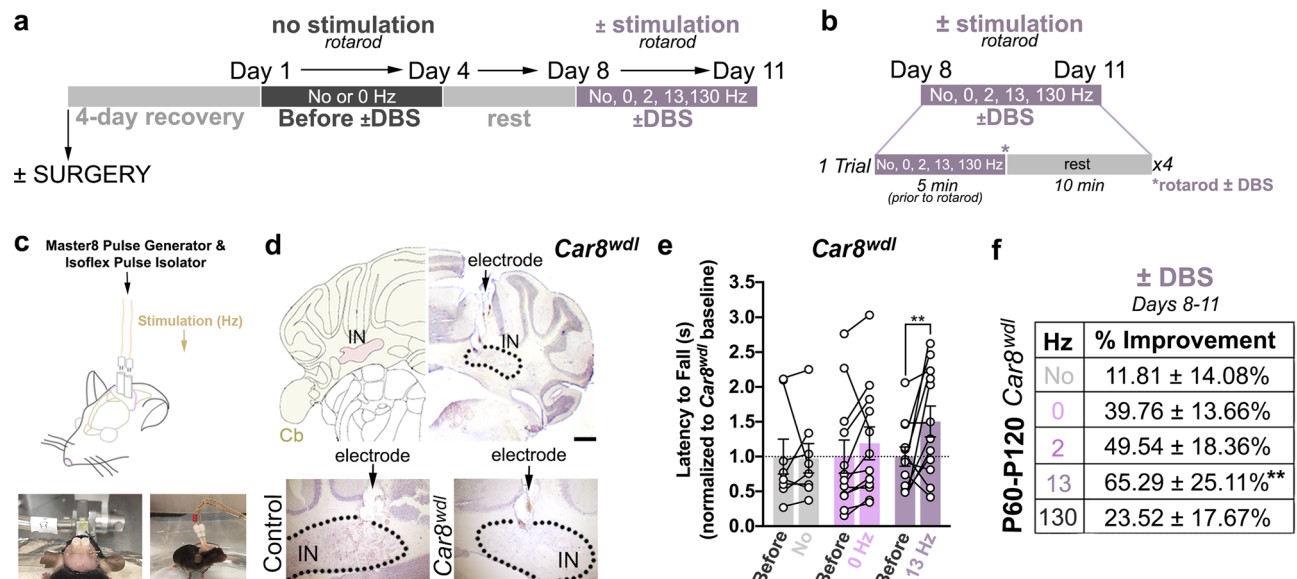

**Fig. 1 13 Hz DBS delivered to the cerebellar interposed nuclei improves the motor behavior of *Car8wdl* mice. a, b** Experimental timeline of the DBS surgery and stimulation paradigm. **c** Schematic and pictures of the DBS surgery and tethered stimulation setup. **d** Example targeting of the DBS electrodes to the interposed nucleus (IN) of control and mutant cerebella (Cb). $n = 6$ biologically independent animals (Control, −DBS: $n = 3$; *Car8wdl*, −DBS: $n = 3$) over 6 independent experiments. The scale bars in (**d**) represent 500 μm (zoomed out) and 50 μm (zoomed in). **e** *Car8wdl* mice that were stimulated at 13 Hz significantly improved on the rotarod ($p = 0.0095$). A black dotted line demarcates the mutant baseline (normalized to $y = 1.00$). The latency to fall values for each *Car8wdl* mouse were plotted, before and during ±DBS. **f** The *Car8wdl* mice that were stimulated at 13 Hz improved the most from their own baselines. $n = 50$ biologically independent animals (No Surgery: $n = 8$; 2 Hz: $n = 6$; 0 Hz: $n = 12$; 13 Hz: $n = 12$; 130 Hz: $n = 12$) over 50 independent experiments. Expanded graphical view can be found in Supplementary Fig. 1. **$p < 0.01$; two-way ANOVA, repeated measures; Sidak's multiple comparisons test; mean ± SEM.

Supplementary Movie 3; No Surgery, 6.58 ± 2.01%, $n = 10$, $p = 0.9463$; 0 Hz, 20.43 ± 7.00%, $n = 15$, $p = 0.0321$; 2 Hz, 28.94 ± 9.57%, $n = 6$, $p = 0.0166$; 13 Hz, 12.12 ± 8.08%, $n = 4$, $p = 0.8414$; 130 Hz, 33.79 ± 8.34%, $n = 11$, $p < 0.0001$. These data show that our cerebellar DBS approach improves motor behavior in *Car8wdl* mice, particularly at a frequency that normally supports proper neural communication of cerebellar output signals.

**Surgical lesions caused by DBS electrodes do not compromise motor circuit integrity.** Cerebellar stimulation has long-distance effects on the motor circuit through modulating cortical excitability[7,35] and corollary discharge[30]. If stimulating the interposed nucleus promotes communication between the cerebellum and the cortex or muscles to improve locomotion, then circuit anatomy should be intact. Therefore, we survey the motor circuit for damage using whole-mount imaging, histology, and antegrade wheat-germ agglutinin (WGA)-Alexa 555 tracing from the cerebellar nuclei (Supplementary Figs. 4–8). While neurodegeneration eliminates key circuit components in the brain, contusions and hemorrhaging impair tissue metabolism and cause massive neuronal depolarization[36,37]. Moreover, glial scarring and inflammation impact neuronal output through stunting axonal regrowth and promoting cell loss, respectively[38,39]. Such effects could counteract cerebellar outflow communication. We found no bruising or hemorrhaging in the cerebellum (Supplementary Fig. 4a), as well as no atrophy or degeneration in vital nodes of the motor circuit after electrode implantation (Supplementary Figs. 5 and 6a). Cerebellar connections are likely preserved in *Car8wdl* mice given that output projections are comparable to that of controls (Supplementary Fig. 7), the critical cell types for motor behavior and learning (namely, Purkinje cells and granule cells) are present (Supplementary Fig. 4b–e; ML thickness (0, 13, 130 Hz): Controls, $n = 3$ (each), 146.812 ± 12.908 μm, 129.146 ± 4.877 μm, 133.463 ± 12.437 μm; Mutants, $n = 3$ (each), 140.788 ± 11.917 μm,

130.628 ± 6.522 μm, 122.363 ± 4.776 μm, $p = 0.9973$ (0 Hz), $p > 0.9999$ (13 Hz), $p = 0.9585$ (130 Hz)), and there is comparable scarring and inflammation (Supplementary Fig. 8; GFAP (0,13, 130 Hz): Controls, $n = 3$ (each), 59.269 ± 6.947, 64.121 ± 5.783, 69.023 ± 7.465; Mutants, $n = 3$ (each), 66.543 ± 3.497, 62.647 ± 5.506, 83.444 ± 11.808, $p > 0.9999$ (0,13 Hz), $p = 0.9556$ (130 Hz); Iba1 (0, 13, 130 Hz): Controls, $n = 3$ (each), 24.609 ± 8.391, 38.690 ± 4.599, 36.339 ± 3.953; Mutants, $n = 3$ (each), 30.630 ± 2.954, 51.615 ± 11.363, 37.827 ± 1.256, $p > 0.9999$ (0, 130 Hz), $p = 0.9486$ (13 Hz)). We also examined brain-muscle integrity by testing muscle responsiveness to cerebellar DBS. We specifically examined the slow-twitch and fast-twitch fiber composition of the tibialis anterior (TA) because previous work demonstrated that it changes after exogenous stimulation[40]. We found that cerebellar stimulation increases slow-twitch fiber protein expression in both control and *Car8wdl* mice, although this change is only statistically significant in the mutant muscles (Supplementary Fig. 9; Controls: 0 Hz: $n = 3$, 10.83 ± 4.59%; 13 Hz: $n = 3$, 31.77 ± 8.73%; $p = 0.0787$; Mutants: 0 Hz, $n = 3$, 22.89 ± 11.13%; 13 Hz: $n = 3$, 47.55 ± 6.22%; $p = 0.0312$). These data show that the *Car8* mutation does not block muscle responsiveness to descending neural modulation. Altogether, we deduce that the surgical lesions created after implanting the DBS electrodes do not impede cerebello-thalamo-cortical communication or the integrity of neuromuscular connectivity. These intact, motor connections may promote functional recovery, which is a key benefit of stimulating *Car8wdl* mice.

**Cerebellar DBS normalizes *Car8wdl* muscle activity during locomotion.** When the cerebello-thalamo-cortical circuit is engaged during locomotion, the muscles are activated in a temporally precise manner[41]. Although direct, exogenous stimulation improves the force, resistance, and length of muscle contractions, it is unclear if neuromodulation alters motor timing to improve

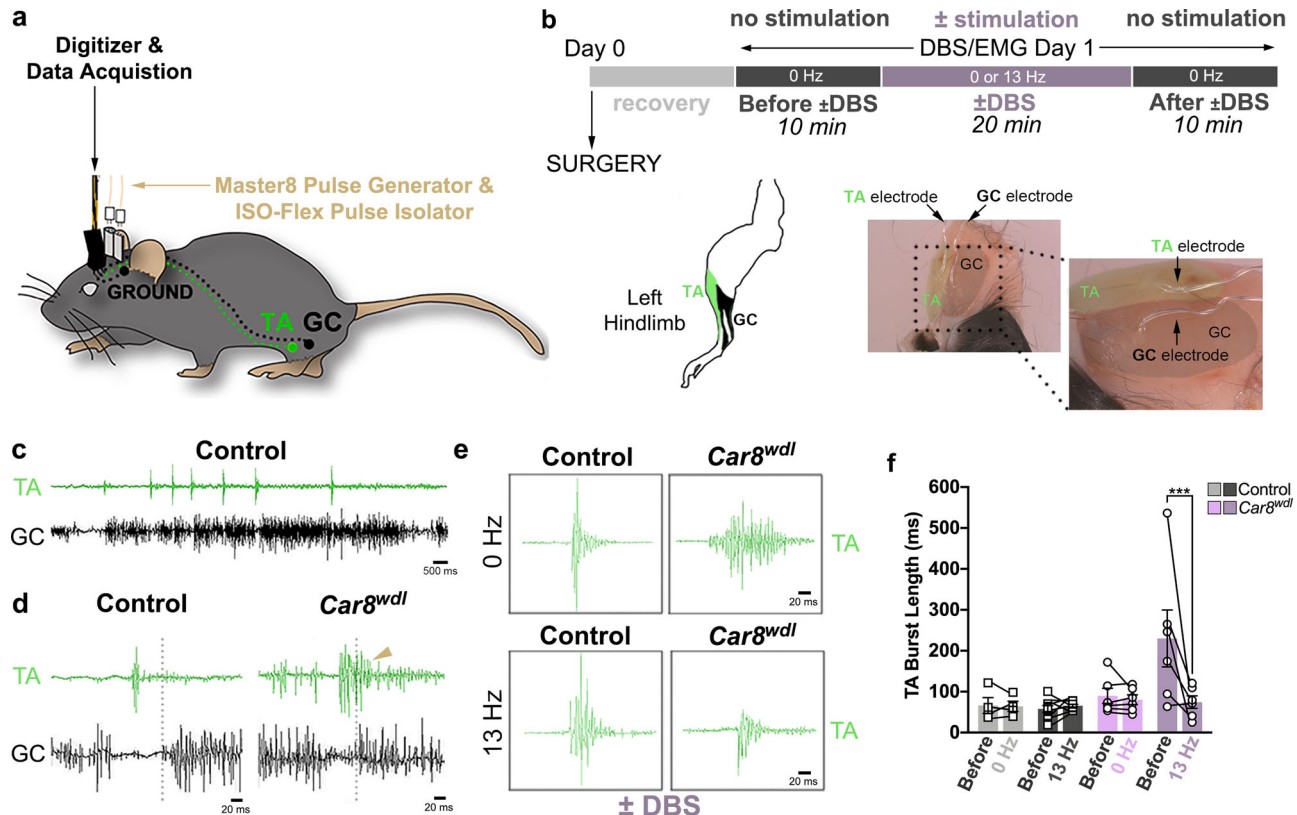

**Fig. 2 13 Hz DBS delivered to the *Car8^wdl* cerebellum improves TA muscle activity. a** Schematic of the surgery that combines DBS with EMG. Mouse was adapted from Miterko et al. (http://creativecommons.org/licenses/by/4.0/)[25]. **b** Experimental timeline of the DBS/EMG surgery and stimulation/recording paradigm. EMG electrodes correctly target the tibialis anterior (TA) and gastrocnemius (GC) hindlimb muscles. **c** An example EMG trace of a control mouse during locomotion. **d** Example EMG traces of control and mutant mice before cerebellar stimulation. Prolonged contractions of the TA muscle occur in *Car8^wdl* mice and causes a co-contraction with the GC muscle. Dotted lines denote the start of a GC contraction. The yellow arrow points to an example of a co-contraction. **e** Examples of TA contractions in non-stimulated and stimulated control and mutant mice. DBS delivered at a frequency of 13 Hz shortens TA contractions in *Car8^wdl* mice. **f** Quantification of TA burst length reveals that 13 Hz DBS significantly shortens the muscle contraction time of *Car8^wdl* mice ($p = 0.0005$). $n = 24$ biologically independent animals (Control: 0 Hz, $n = 4$, 13 Hz, $n = 8$; *Car8^wdl*: 0 Hz, $n = 6$, 13 Hz, $n = 6$) over 24 independent experiments. ***$p < 0.001$; three-way ANOVA, repeated measures; Sidak's multiple comparisons test; mean ± SEM.

locomotion[42]. To test this, we measured electromyography (EMG) activity in active *Car8^wdl* mutant mice (Fig. 2a, b). We previously showed that *Car8^wdl* mice have prolonged contractions of the tibialis anterior (TA) muscle[25]. The TA is important for locomotion because this task requires its activity to fire out-of-phase with the activity of the gastrocnemius muscle (GC; Fig. 2c, d). When the activities of the TA and GC muscles overlap, co-contractions occur, and locomotion is impaired (Fig. 2d)[25,43]. In *Car8^wdl* mice, prolonged TA firing likely contributes to their locomotor deficits by increasing the probability that their hindlimb muscles co-contract (Fig. 2d)[25]. However, this abnormal muscle phenotype is rescued in *Car8^wdl* mice, when EMG is combined with 13 Hz cerebellar DBS (Fig. 2e, f; $n = 6$; 74.42 ± 15.22 ms; $p = 0.0005$). We conducted several control experiments to support this result. Without DBS, the TA contracts for longer in *Car8^wdl* mice than in the control mice (Fig. 2e; Supplementary Data 1; Controls: 63.77 ± 7.99 ms, $n = 19$; Mutants: 147.40 ± 26.51 ms, $n = 19$; $p = 0.0046$)[25]. Stimulating the *Car8^wdl* cerebellum at 2 and 130 Hz does not shorten TA contractions (Supplementary Fig. 10; 2 Hz: 106.98 ± 34.38 ms, $n = 3$, $p = 0.9731$; 130 Hz: 201.53 ± 45.02 ms, $n = 4$, $p = 0.9052$). No stimulation frequency affects TA contraction times in control mice (Supplementary Fig. 6; 0 Hz: 64.01 ± 12.52 ms, $n = 4$, $p = 0.9995$; 2 Hz: 46.35 ± 16.15 ms, $n = 3$, $p = 0.9718$; 13 Hz: 65.78 ± 3.37 ms, $n = 8$, $p = 0.8007$; 130 Hz: 67.76 ± 13.11 ms, $n = 4$, $p = 0.0646$). Interestingly, the improvement we find in mutant TA firing is specific to this muscle

property. For instance, we find that the *Car8^wdl* TA contracts as often with 13 Hz DBS (0.817 ± 0.357 bursts/ms) as it does without it (Supplementary Fig. 10; $n = 6$, 0.520 ± 0.114 bursts/ms, $p = 0.6495$). Therefore, DBS likely restores, rather than compensates for, *Car8^wdl* muscle behavior. Our data reveal that cerebellar DBS evokes widespread responses throughout the motor system. With the data presented in Fig. 1, the results also suggest that cerebellar nuclei DBS may improve locomotion in part by normalizing the function of voluntary muscles in *Car8^wdl* mice.

**Success of low-frequency cerebellar DBS depends on multiple motor behavior responses**. We next tested how neuromodulation impacts mobility during different motor tasks. We measured the effects of DBS on the general locomotion of *Car8^wdl* mice by using the open field assay, and we studied the repercussions of stopping DBS on muscle activity and motor learning, through using EMG and the accelerating rotarod, respectively. When combining cerebellar DBS with the open field assay, we found that only 13 Hz prevents motor degradation after surgery in *Car8^wdl* mice (Fig. 3a, b; Supplementary Fig. 11; No Surgery, $n = 3$, Day 0, 1276.00 ± 141.37 s versus Day 13, 1162.00 ± 157.77 s, $p = 0.7089$; 0 Hz, $n = 5$; Day 0, 1255.66 ± 147.55 s versus Day 13, 793.26 ± 109.66 s, $p = 0.0017$; 13 Hz: $n = 3$, Day 0, 1297.87 ± 78.92 s versus Day 13, 1268.90 ± 19.77 s, $p = 0.9771$; No Surgery/13 Hz, Day 13, $p = 0.5382$). Because 13 Hz DBS had the most

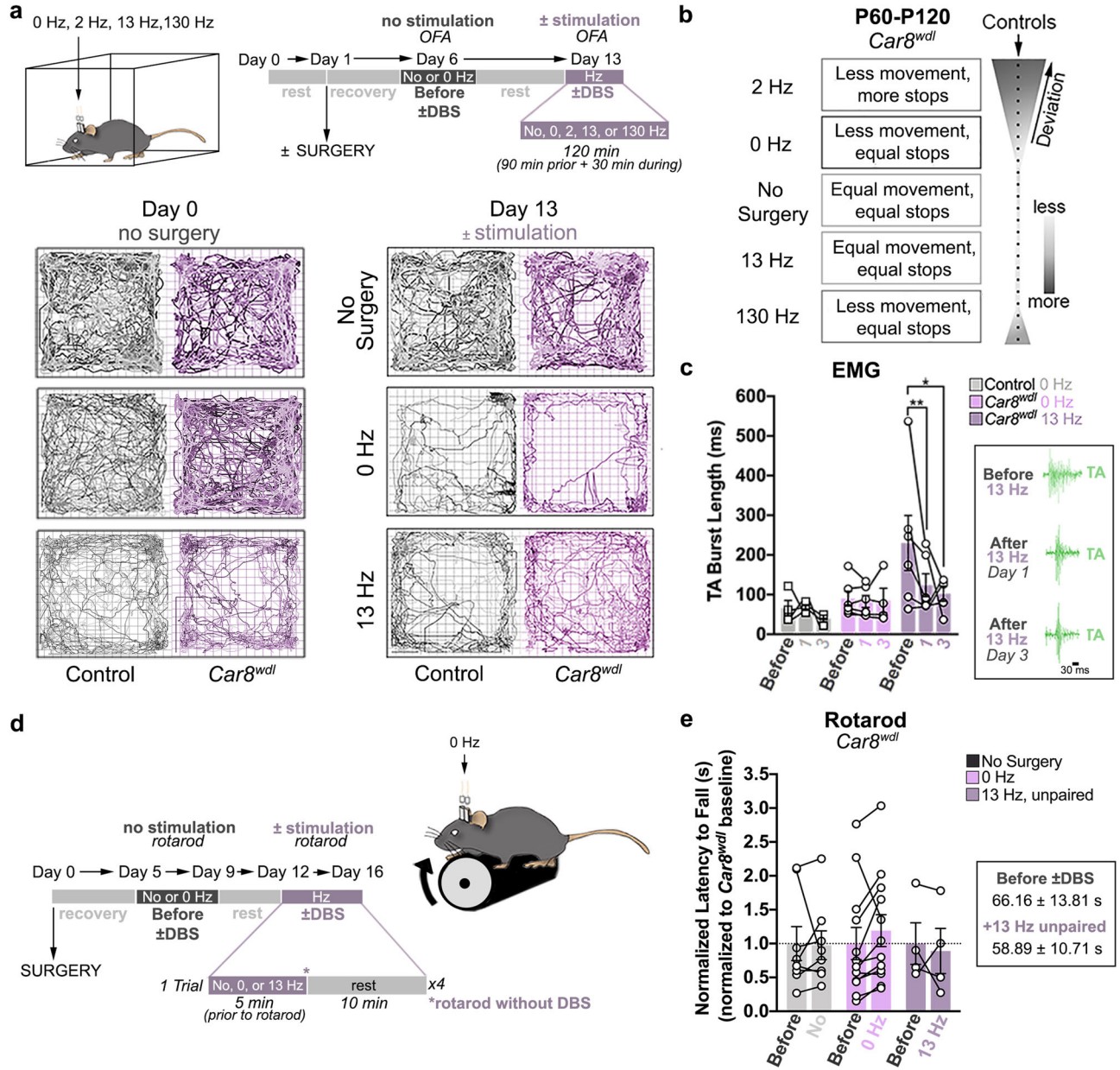

**Fig. 3 Cerebellar DBS promotes locomotor improvements for ongoing motor function. a** Schematic and experimental timeline of the open field assay (OFA) used in combination with DBS to measure movement and movement time. Example traces reveal that 13 Hz DBS maintains the movement of Car8$^{wdl}$ mice. **b** No other stimulation frequency other than 13 Hz preserves the movement of Car8$^{wdl}$ mice. Schematic summarizes the effects of cerebellar stimulation on Car8$^{wdl}$ movement and compares how alterations cause Car8$^{wdl}$ behavior to deviate from that of controls. The darker the gray, the more stimulation impacted Car8$^{wdl}$ movement. The wider the schematic, the more Car8$^{wdl}$ motor behavior deviated from control motor behavior (black dotted vertical line). The schematic aligns with the stimulation frequencies on the left. **c** TA muscle firing in Car8$^{wdl}$ mice remains reduced after stopping DBS (Day 1: $p = 0.0013$; Day 3: $p = 0.0492$). Example traces from a 13-Hz stimulated Car8$^{wdl}$ mouse are outlined. Before = Before ±DBS, Day 1; 1 = After ±DBS, Day 1; 3 = After ±DBS, Day 3. $n = 24$ biologically independent animals (Control, 0 Hz: $n = 4$; Control, 13 Hz: $n = 8$; Car8$^{wdl}$, 0 Hz: $n = 6$; Car8$^{wdl}$, 13 Hz: $n = 6$) over 24 independent experiments on Day 1. $n = 15$ biologically independent animals (Control, 0 Hz: $n = 3$; Control, 13 Hz: $n = 3$; Car8$^{wdl}$, 0 Hz: $n = 4$; Car8$^{wdl}$, 13 Hz: $n = 5$) over 15 independent experiments on Day 3. **d** Experimental timeline and schematic of our unpaired behavioral paradigm. The mutants perform on the rotarod without DBS. **e** Car8$^{wdl}$ mice need to be stimulated while on the rotarod to receive benefits ($p = 0.9999$). The black dotted line demarcates the mutant baseline (normalized to $y = 1.00$). The latency to fall values for each Car8$^{wdl}$ mouse were plotted, before and during unpaired DBS. $n = 24$ biologically independent animals (No Surgery: $n = 8$; 0 Hz: $n = 12$; unpaired 13 Hz: $n = 4$) over 24 independent experiments. $*p < 0.05$; $**p < 0.01$; three-way ANOVA, repeated measures with mixed-effects (**c**); two-way ANOVA, repeated measures (**e**); Sidak's multiple comparisons test (**c**, **e**); mean ± SEM.

beneficial effects on muscle function (Fig. 2) and the least negative impact on freely moving behavior (Fig. 3a, b), these data might explain why only 13-Hz stimulated Car8$^{wdl}$ mice improve their performances on the rotarod (Fig. 1).

To test whether 13 Hz DBS promotes locomotion by normalizing Car8$^{wdl}$ muscle function, we measured muscle activity after stopping DBS (Fig. 3c). Interestingly, we found that Car8$^{wdl}$ TA function remains corrected (Fig. 3c; $n = 6$; $123.79 \pm 28.80$ s;

Before/After, $p = 0.0013$). Stimulating the cerebellum daily, as performed in the rotarod experiments, further suppresses TA pathology (Supplementary Fig. 12; $n = 5$; 102.32 ± 19.27 ms; Day 1/Day 3, $p = 0.0492$). If normalized muscle function alone mediates improvements, then we would expect $Car8^{wdl}$ mice to achieve the same benefits, even when DBS is no longer being delivered during the behavior. To test this possibility, we stimulated $Car8^{wdl}$ mice, then terminated the DBS right before placing the mice onto the rotarod (Fig. 3d). $Car8^{wdl}$ mice did not improve using this paradigm (Fig. 3e; 13 Hz, unpaired: $n = 4$, −4.39 ± 31.42%, $p = 0.9999$), which suggests that the ideal context for improving complex motor behaviors, such as motor coordination, in $Car8^{wdl}$ mice requires 13 Hz DBS to engage the motor function machinery that is actively operating at the time of the stimulation.

**Cerebellar DBS induces long-lasting motor benefits in $Car8^{wdl}$ mice.** One way in which ongoing learning manifests is through gait adaptations. By combining DBS with footprint and DigiGait analyses, we found that 13 Hz DBS alters the step cycle, but not the overall gait of $Car8^{wdl}$ mice (Fig. 4a–e; Supplementary Fig. 13). A step cycle consists of braking, propulsion, and a swing phase. The paw contacts the ground during "braking" and "propulsion," but not during the "swing" (Fig. 4b). The mice experience the greatest stability while braking. With reference to these measures, $Car8^{wdl}$ mice spend significantly less time braking ($n = 21$; 0.094 ± 0.004 s) than do control mice (Fig. 4c; $n = 19$, 0.110 ± 0.004 s, $p = 0.0037$), which corroborate our previous findings that the paws of $Car8^{wdl}$ mice contact the ground less during locomotion than do the paws of control mice[25]. However, 13 Hz DBS prolongs the braking of $Car8^{wdl}$ hindlimbs (Fig. 4e; Day 13: 0 Hz, $n = 5$, 0.093 ± 0.010 s, $p = 0.9498$; 13 Hz, $n = 4$, 0.110 ± 0.002 s, $p = 0.00307$). Interestingly, the response is sustained 7 days after stimulation stops (Fig. 4e; Day 20: 0 Hz, $n = 5$, 0.082 ± 0.003 s, $p = 0.9484$; 13 Hz, $n = 4$, 0.114 ± 0.013 s, $p = 0.1034$). Compared to control mice ($n = 4$), these results reveal that DBS corrects the mutant step cycle (Fig. 4e; Supplementary Data 1; Day 0: 0.127 ± 0.008 s; Day 20: 0.124 ± 0.006 s, $p = 0.9993$; Control/Mutant (+13 Hz): Day 0, $p = 0.0076$ versus Day 20, $p = 0.8727$) by stabilizing ongoing movements in $Car8^{wdl}$ mice, during and after stimulation.

To investigate whether other motor improvements are retained, we tracked the movement of $Car8^{wdl}$ mice and measured their coordination, 4–7 days after DBS was introduced (Fig. 4f–h; Rotarod, Days 15–18, Open field, Day 20). We found that previously stimulated $Car8^{wdl}$ mice continue to exhibit increased locomotion (Fig. 4g, h; No Surgery, $n = 3$, Day 20, 960.63 ± 131.53 s, $p = 0.1098$; 0 Hz, $n = 5$, Day 20, 892.74 ± 129.93 s, $p = 0.0134$; 13 Hz: $n = 3$, Day 20, 975.43 ± 125.77 s, $p = 0.1008$) and they show improvements on the rotarod, starting 4 days after stimulation stops (Fig. 4f; Supplementary Movie 4; No Surgery, $n = 8$, 33.62 ± 21.85%, $p = 0.6413$; 0 Hz, $n = 5$; 31.01 ± 34.26%, $p = 0.4267$; 13 Hz, $n = 12$, 120.14 ± 34.12%, $p = 0.0055$). These results underscore the long-term benefits of providing 13 Hz cerebellar DBS to restore mobility in $Car8^{wdl}$ mutant mice.

**Ataxia severity contributes to the variability of DBS outcomes.** We show that stimulating cerebellar outputs has therapeutic value, but the benefits are variable, as reported in humans[44,45]. Besides inefficient electrode targeting[46] and insertional damage[47], inappropriate stimulus parameters[48] and suboptimal electrode design and impedance[49,50] also impact DBS effectiveness. Despite efforts to reduce the impact of these factors on patient outcomes[51–53], responses still vary due to differences in age[54] and disease severity[55]. A retrospective analysis of our data revealed

that 58.3% ($n = 7/12$) of the $Car8^{wdl}$ mice responded favorably to 13 Hz DBS (Fig. 5a; we defined mice as "responders" to DBS if they improved more than that of sham mice (39.76 ± 13.66%) on the rotarod during stimulation). A previous publication showed that a range of beta-frequencies (10, 20, and 30 Hz) comparably improves the fall rate of rodents[31]. However, higher beta-frequencies (20–30 Hz) more effectively suppress incoordination by enhancing cerebello-thalamo-cortical communication[31,35,56]. We therefore tested whether increasing the stimulation frequency to 20 Hz would improve DBS responses, but found that the same number of $Car8^{wdl}$ mice improve with 20 Hz DBS (57.1%; $n = 8/14$) and with equal robustness (Supplementary Fig. 14; $p = 0.5660$), as compared to 13 Hz DBS. This shows that multiple beta-frequencies are effective at improving motor coordination in $Car8^{wdl}$ mice. Moreover, the data indicate that the therapeutic benefits of cerebellar DBS encompass an optimal range of frequencies rather than a single frequency. We therefore predict that there is flexibility in what might be considered an optimal therapy, which depends on each individual's inherent circuit properties.

Another possibility for the observed variability may be symptom severity before stimulation. Indeed, when we plotted the latency to fall values for the mutant responders ($n = 7$, 13 Hz; $n = 8$, 20 Hz), we observed that stimulated $Car8^{wdl}$ mice performed the best when their ataxia was less severe (Fig. 5b; Supplementary Fig. 14c). If ataxia severity impacts responsiveness (measured as motor improvements on the rotarod), then DBS should improve motor function in $Car8^{wdl}$ mice whose ataxia has not progressed too far beyond the onset of symptoms at the time of stimulation. As hypothesized, we found that beta-frequency DBS was therapeutic for $Car8^{wdl}$ mice with less severe ataxia (Fig. 5c; P30; 0 Hz: $n = 4$, 47.15 ± 9.97%, $p = 0.0840$; 13 Hz: $n = 4$, 130.05 ± 53.21%, $p = 0.0076$) and DBS did not worsen movement or gait (Supplementary Fig. 15; Supplementary Data 1). Importantly, we found that stimulating $Car8^{wdl}$ mice at the peak of their ataxia was not beneficial (Fig. 5d; Supplementary Fig. 16; ≥P150; Rotarod, 0 Hz: $n = 6$, 114.23 ± 42.80%, $p = 0.2980$; 13 Hz: $n = 5$, 87.20 ± 82.24%, $p = 0.5557$; EMG, $n = 3$, 96.39 ± 33.92 ms (Before DBS); 180.24 ± 51.89 ms (+13 Hz), $p = 0.4208$; 261.49 ± 66.94 ms (After DBS), $p = 0.0465$). Together, these data indicate that a positive behavioral outcome to cerebellar DBS partially depends on the severity of ataxia at the time when treatment is initiated.

**Purkinje cell firing properties change with motor deterioration in $Car8^{wdl}$ mice.** Given that $Car8^{wdl}$ responders continue to improve after stimulation stops and that ataxia severity restricts DBS efficacy, circuit properties may help set the therapeutic window. We hypothesized that Purkinje cell activity may specifically impact DBS effectiveness due to their roles in motor learning and ataxia in $Car8^{wdl}$ mice[24,25]. Although it is unclear which Purkinje cell properties cause ataxia, alterations in their firing regularity disrupt cerebellar nuclei output and motor function[57,58]. Contributing to the motor dysfunction of $Car8^{wdl}$ mice is erratically firing Purkinje cells (i.e., increased pausing, increased coefficient of variation, CV) with reduced spike time variability (i.e., decreased CV2)[24]. While some of these features (i.e., decreased CV2) manifest during development (~P20), when ataxia starts[25], others (i.e., increased pausing, increased CV) emerge in $Car8^{wdl}$ mice (P60–P90)[24] when motor function worsens (Supplementary Fig. 2). If motor deficits correlate with the regularity of Purkinje cell firing, then alterations in pausing, CV, and CV2 may also impact DBS effectiveness. We therefore measured the regularity of Purkinje cell firing in $Car8^{wdl}$ mice that respond (P30) and do not respond (≥P150) to DBS, through performing in vivo extracellular recordings (Fig. 6a, b). $Car8^{wdl}$

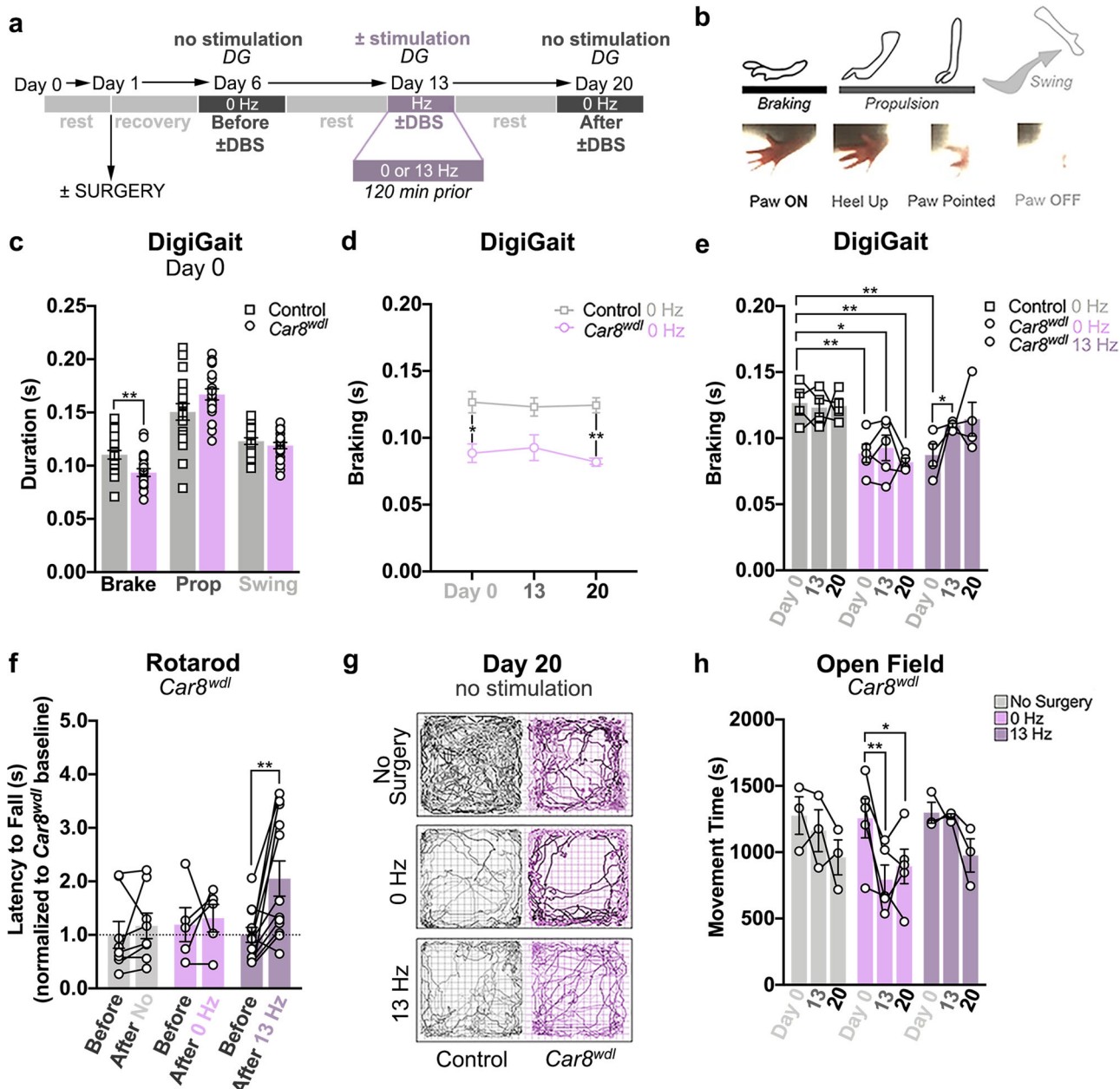

**Fig. 4 *Car8^wdl* motor gains are long-lasting. a** Experimental timeline where DBS was combined with DigiGait (DG) analysis. **b** A detailed schematic of the different phases of the step cycle. **c** *Car8^wdl* mice spend significantly less time braking than do the control mice ($p = 0.0108$). $n = 40$ biologically independent animals (Control: $n = 19$; *Car8^wdl*: $n = 21$) over 40 independent experiments. **d** The braking deficits of non-stimulated *Car8^wdl* mice are sustained over time (Day 0: $p = 0.0286$; Day 13: $p = 0.1075$; Day 20: $p = 0.0048$). $n = 9$ biologically independent animals (Control, 0 Hz: $n = 4$; *Car8^wdl*, 0 Hz: $n = 5$) over 9 independent experiments. **e** 13 Hz cerebellar DBS prolongs the braking time of *Car8^wdl* mice ($p = 0.0307$). $n = 17$ biologically independent animals (Control, 0 Hz: $n = 4$; Control, 13 Hz: $n = 4$; *Car8^wdl*, 0 Hz: $n = 5$; *Car8^wdl*, 13 Hz: $n = 4$) over 17 independent experiments. **f** *Car8^wdl* mice that were stimulated at 13 Hz continue to improve on the rotarod 4–7 days after stimulation has been stopped ($p = 0.0055$). The black dotted line demarcates the mutant baseline (normalized to $y = 1.00$). The latency to fall values for each *Car8^wdl* mouse were plotted, before and after ±DBS. $n = 25$ biologically independent animals (No Surgery: $n = 8$; 0 Hz: $n = 5$; 13 Hz: $n = 12$) over 25 independent experiments. Expanded graphical views can be found in Supplementary Fig. 14. **g** Representative open field traces that show the prolonged effects of 13 Hz DBS on *Car8^wdl* locomotion. **h** *Car8^wdl* mobility remains deteriorated with 0 Hz (Day 20: $p = 0.0134$), but not 13 Hz DBS (Day 20: $p = 0.1008$). $n = 11$ biologically independent animals (No Surgery: $n = 3$; 0 Hz: $n = 5$; 13 Hz: $n = 3$) over 11 independent experiments. Expanded graphical views can be found in Supplementary Fig. 11. *$p < 0.05$; **$p < 0.01$; unpaired, two-tailed Student's *t*-test (**c**); three-way ANOVA, repeated measures with mixed-effects (**e**); two-way ANOVA, repeated measures (**h**) with mixed-effects (**d**, **f**); Sidak's multiple comparisons test (**d**, **e**); Dunnett's multiple comparisons test (**f–h**); mean ± SEM.

Purkinje cells only fire more irregularly on the whole than controls when motor function has significantly deteriorated (Fig. 6c–f; P30 (pause percent, CV): Control, $n = 25$ cells from 4 mice, $23.78 \pm 2.98\%$, $3.864 \pm 0.337$ versus Mutant, $n = 14$ cells from 3 mice, $29.65 \pm 5.21\%$, $4.682 \pm 0.576$, $p = 0.2981$, $p = 0.1969$;

≥P150 (pause percent, CV): Control, $n = 11$ cells from 4 mice, $2.07 \pm 1.07\%$, $0.757 \pm 0.188$ versus Mutant, $n = 11$ cells from 7 mice, $18.08 \pm 4.13\%$, $4.597 \pm 1.169$, $p = 0.0013$, $p = 0.0041$). However, Purkinje cell firing is more regular during local spike trains in *Car8^wdl* mice at P20–P90[24,25], but not at ≥P150

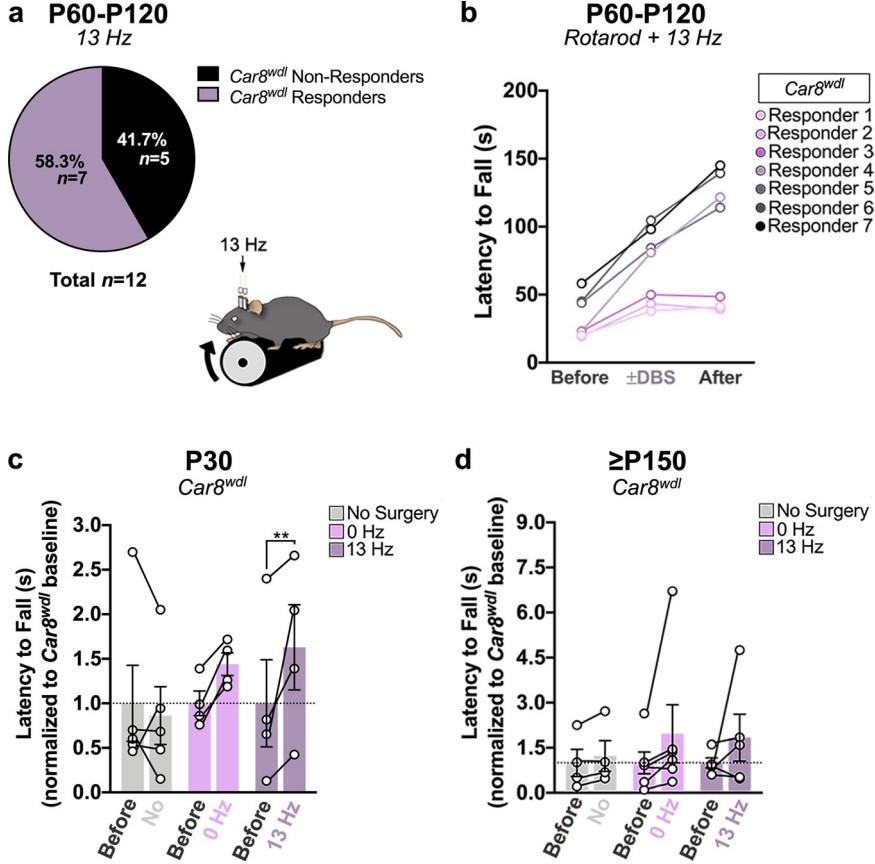

**Fig. 5 Ataxia severity contributes to the DBS outcomes of Car8$^{wdl}$ mice. a** 58.3% of stimulated Car8$^{wdl}$ mice (n = 7/12) respond to 13 Hz DBS.
**b** Improvements depend on the motor function of Car8$^{wdl}$ mice prior to being stimulated at 13 Hz. Each line represents a Car8$^{wdl}$ "responder" animal (n = 7). **c** On average, P30 Car8$^{wdl}$ mice improve with 13 Hz stimulation (p = 0.0076). n = 13 biologically independent animals (No Surgery: n = 5; 0 Hz: n = 4; 13 Hz: n = 4) over 13 independent experiments. **d** On average, ≥P150 Car8$^{wdl}$ mice do not improve with 13 Hz stimulation (p = 0.5557). n = 15 biologically independent animals (No Surgery: n = 4; 0 Hz: n = 6; 13 Hz: n = 5) over 15 independent experiments. **p < 0.01; two-way ANOVA, repeated measures (**c**, **d**); Sidak's multiple comparisons test (**c**, **d**); mean ± SEM (**c**, **d**).

(Fig. 6c–f; P30 (CV2): Control, n = 25 cells from 4 mice, 0.394 ± 0.023 versus Mutant, n = 14 cells from 3 mice, 0.214 ± 0.025, p < 0.0001; ≥P150 (CV2): Control, n = 11 cells from 4 mice, 0.405 ± 0.034 versus Mutant, n = 11 cells from 7 mice, 0.320 ± 0.028, p = 0.0921). This could be problematic because local variation in spike trains transmits motor information to the cerebellar nuclei[59] and controls behavior[60,61]. Also, because ataxia improves after correcting CV2 in Car8$^{wdl}$ mice[24], DBS responsiveness may require highly regular Purkinje cell signals to be present. In summary, our electrophysiology data show that Purkinje cell firing patterns are differentially decoded by the downstream nuclei cells during the progression of ataxia and that spike timing in Purkinje cells may need to be modified prior to stimulation in order for the more severe Car8$^{wdl}$ mice to improve. The data also raise the intriguing possibility that although DBS likely targets circuits within a given area, there may be critical nodes, such as the Purkinje cell-to-cerebellar nuclei connection, that must be intact for the DBS to work best.

**Cerebellar DBS relies on Purkinje cell neurotransmission.**
Because we found that Purkinje cell function, motor behavior, and DBS responsiveness are inextricably linked, we asked whether modulating Purkinje cell neurotransmission is vital for treating ataxia. Previous attempts at improving ataxia through the use of pharmacologically-based treatments reveal that the greatest therapeutic effects emerge after normalizing Purkinje cell function,

regardless of the model or genotype studied[10,24,62]. Therefore, we postulated that stimulating L7$^{Cre}$;Vgat$^{flox/flox}$ mice[63], a second ataxia model which lacks Purkinje cell GABAergic neurotransmission (Fig. 6g), would result in no improvements. The rationale for using this model is that Purkinje cell output is selectively blocked, a manipulation that precisely targets functional circuit properties without inducing large-scale anatomical rearrangements or neurodegeneration[63]. As hypothesized, cerebellar DBS did not restore proper motor function in L7$^{Cre}$;Vgat$^{flox/flox}$ mice (Fig. 6h, i; No Surgery: n = 3, −1.66 ± 6.37%, p = 0.9989; 0 Hz: n = 5, 27.81 ± 25.55%, p > 0.9999; 20 Hz: n = 5, −4.52 ± 17.96%, p = 0.9927). Given that eliminating GABAergic neurotransmission from Purkinje cells alone produces diverse motor deficits, such as disequilibrium and postural problems in addition to incoordination, it may be that different modes of chemical or electrical manipulations would fail to elicit improvements for multiple cerebellar deficits. Indeed, we recently demonstrated that administering a tremorgenic drug to L7$^{Cre}$;Vgat$^{flox/flox}$ mice does not alter motor behavior either[27]. These results indicate that a closed-loop cerebellar circuit is needed to propagate or rescue motor dysfunction in ataxic mice. Therefore, the therapeutic signals provided by the cerebellar nuclei DBS could conceivably exit the cerebellum and then return to use the computational power of the Purkinje cells to mediate circuit repair, or the signals could retrogradely travel from the cerebellar nuclei back into the cerebellar cortex to more directly impact Purkinje cell function.

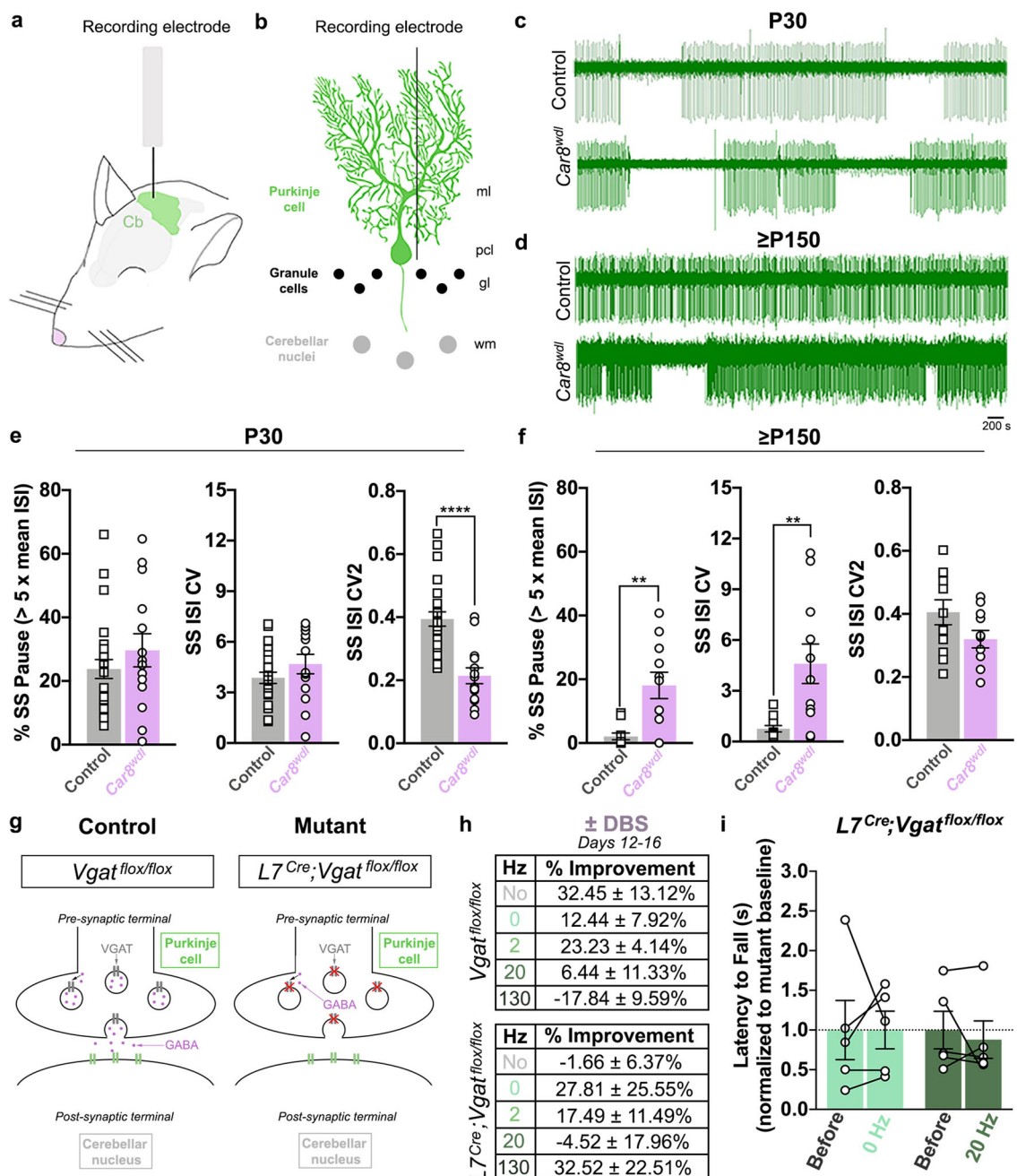

**Fig. 6 Cerebellar DBS requires Purkinje cell neurotransmission to be effective. a** Schematic of our in vivo electrophysiology, anesthetized setup. Cb (green) = cerebellum. **b** Schematic showing the extracellular recording of a Purkinje cell. ml = molecular layer; pcl = Purkinje cell layer; gl = granule layer; wm = white matter. **c, d** Example traces of Purkinje cell activity from P30 (**c**) and ≥P150 (**d**) control and Car8^wdl mice. **e** P30 Car8^wdl Purkinje cells fire as irregularly as age-matched control cells, on the whole ($p = 0.1969$), but more regularly during local trains ($p < 0.0001$). $n = 39$ cells (Control: $n = 25$; Car8^wdl: $n = 14$) from 7 biologically independent animals (Control: $n = 4$; Car8^wdl: $n = 3$) over 7 independent experiments. **f** Aged Car8^wdl Purkinje cells fire more irregularly than do age-matched control cells, on the whole ($p = 0.0041$), and equally as regular during local trains ($p = 0.0921$). $n = 22$ cells (Control: $n = 11$; Car8^wdl: $n = 11$) from 11 biologically independent animals (Control: $n = 4$; Car8^wdl: $n = 7$) over 11 independent experiments. **g** Schematic detailing how Purkinje cell neurotransmission is blocked in L7^Cre;Vgat^flox/flox mice. Adapted from Brown et al. (https://creativecommons.org/licenses/by/4.0/)[27]. **h** Cerebellar DBS requires Purkinje cell activity to mediate improvements to motor function in L7^Cre;Vgat^flox/flox mice. **i** Normalized latency to fall values showing that L7^Cre;Vgat^flox/flox mice do not improve with 20 Hz DBS ($p = 0.9927$). The black dotted line demarcates the mutant baseline (normalized to $y = 1.00$). $n = 10$ biologically independent animals (0 Hz: $n = 5$; 20 Hz: $n = 5$) over 10 independent experiments. **$p < 0.01$; ****$p < 0.0001$; unpaired, two-tailed Student's $t$-test (**e, f**); two-way ANOVA, repeated measures (**i**); Sidak's multiple comparisons test (**i**); mean ± SEM (**e, f, i**).

## Discussion

We show that stimulating the Car8^wdl cerebellum at 13 Hz results in short-term and long-term motor improvements. While neurostimulation improves muscle function (Fig. 2) and general mobility in Car8^wdl mice (Fig. 3), improvements to motor coordination (Fig. 1) and stepping (Fig. 4) require DBS to continue during exercise (Fig. 3). These data highlight the importance of treatment design for DBS effectiveness: not only does electrode

targeting matter, but so does delivery. Since coordinative treadmill training improves ataxia[64,65], as well as Purkinje cell and muscle health[66–68], we hypothesize that supplementing DBS with skilled exercise improves cerebellar learning and muscle function in order to facilitate retention and enhance motor recovery.

We provide several lines of evidence showing that DBS improves cerebellar and muscle functions, including that stimulation improves motor coordination and muscle function during movement (Figs. 1 and 2; Supplementary Movie 1). These results can partly be attributed to our targeting of the interposed cerebellar nuclei (Supplementary Fig. 1). The interposed cerebellar nuclei communicate with neurons in the thalamus, red nucleus, and spinal cord to control ongoing movements. On the local circuit level, the interposed nucleus facilitates motor learning[8]. Although we cannot rule out the possibility that the dentate and fastigial nuclei are also stimulated in our paradigm given the relatively small size of the interposed, our rotarod, DigiGait, and EMG findings support predominant interposed stimulation for at least two reasons: sensorimotor feedback likely contributes to $Car8^{wdl}$ motor improvements and locomotor adaptation occurs. The interposed nucleus specifically receives sensory information from spinal circuits to modulate locomotion[28]. In our study, cerebellar DBS improves complex motor behaviors in $Car8^{wdl}$ mice only when it is paired with tasks that generate abundant sensorimotor feedback, such as the rotarod (Fig. 3). Our EMG data also corroborate the role of sensorimotor feedback in mediating DBS responses. Not only does sensory feedback modulate the EMG activity of locomotor muscles[69], it also alters motor function in a predictable manner: first temporally, then spatially[28]. In our study, we found that DBS improves TA firing in as little as 20 min (Fig. 2), but behavioral modifications occur anywhere from 2 h (Fig. 3) to 3–18 days later (Fig. 4; Supplementary Figs. 11 and 14).

Besides reinforcing our targeting, our behavioral data provides insight into how neuromodulation might permanently restore locomotion in vivo. With regards to muscle function, our open field and EMG results support the possibility of DBS altering the function of multiple muscle groups. Here, we studied the TA given its dysfunction in $Car8^{wdl}$ mice[25], its role in motor precision[70], and its ability to respond to exogenous electrical stimulation[40]. However, other locomotor muscles respond to electrical currents, including the soleus, which is activated whenever mice engage in slow, postural movements, such as standing and walking[71]. In contrast, the GC and TA are activated during fast, coordinated movements, such as running and jumping, or when bending and positioning the paws[71]. Because 13 Hz DBS rescues movement in $Car8^{wdl}$ mice during a task that does not require perfect coordination or rapid mobility (i.e., open field), it is likely that DBS also activates muscles such as the soleus to restore motor function. Accordingly, since 13 Hz DBS does not alter how often the TA contracts (Supplementary Fig. 10), nor do the other stimulation frequencies, changes in TA activity alone cannot explain all the parameters of $Car8^{wdl}$ movement after surgery.

With regards to motor circuit function, 13 Hz DBS likely engages cerebello-thalamo-cortical pathways given that coordination and gait kinematics improve with stimulation. Cerebello-thalamo-cortical pathways communicate at 13 Hz and regions included in this circuit (cerebellum, basal ganglia, thalamus, cortex, and spinal cord) control paw placement, contact area, and coordination[34,72]. Other frequencies, including 20 and 100 Hz, also accomplish this, albeit to different degrees. While stimulating the cerebellum at 20 and 100 Hz enhances cortical excitability, the effects are maximized and sustained only at 20 Hz[35]. This reveals that boosting motor signal transmission is frequency-dependent, with the most optimal effects achieved within the beta-band. Our data corroborate these findings by showing that $Car8^{wdl}$ mice

similarly improve with 13 and 20 Hz DBS (Supplementary Fig. 14), but not with higher frequencies (Supplementary Fig. 1). Our finding that control mice benefit from 0, 2, and 130 Hz, but not 13 Hz, reveals that there may be a ceiling effect for how much control mice can improve (Supplementary Fig. 3). This is evidenced by our histology data, which shows comparable changes to muscle composition between the control and $Car8^{wdl}$ mice after providing 13 Hz DBS (Supplementary Fig. 9; 13-Hz stimulated control compared to 13-Hz stimulated $Car8^{wdl}$, Slow: $p = 0.5813$), and our rotarod data, which shows that the motor improvements in control mice depend on an initial presence of behavioral deficits—e.g., due to implant lesions. For example, the control mice with significantly worse (0 and 130 Hz) or highly variable (2 Hz) starts, are the groups that improve with DBS (Supplementary Fig. 3; No Surgery, $n = 10$, $267.313 \pm 5.601$ s; 0 Hz, $n = 15$, $194.358 \pm 15.628$ s; 2 Hz, $n = 6$, $200.757 \pm 18.544$ s; 130 Hz, $n = 11$, $187.591 \pm 15.147$ s; No/0 Hz, $p = 0.0076$; No/2 Hz, $p = 0.1410$; No/130 Hz, $p = 0.0058$). After DBS has been initiated, 0, 2, and 130-Hz stimulated control mice perform equally as well on the rotarod as any other group.

Perhaps most insightful toward understanding how DBS works in vivo are the long-lasting benefits that $Car8^{wdl}$ mice receive after stimulation (Figs. 3–5), as this suggests that carryover effects are maintained by motor function circuitry. In humans, carryover effects have been reported after stimulating the basal ganglia in dystonia and Parkinson's disease patients[22,23]. Symptom dissolution (during DBS) and re-emergence (after DBS) typically follow the same timescale. The symptoms that DBS first reduces —e.g., tremor and repetitive movements—are also the first to return[22,23]. Likewise, the symptoms that DBS corrects last—e.g., locomotion and sustained spasticity—return last[22,23]. Patterns of reoccurrence temporally differ because DBS alters different cellular processes, including the desynchronization of neural activity (which occurs instantaneously) and the induction of long-term potentiation (which occurs after the circuit adapts)[23,73]. In our ataxia model, improvements do not dissipate after 1 week, which indicates that stimulating the cerebellum may remodel motor circuit connectivity[73,74] or harness the computational power and plasticity of Purkinje cells. Indeed, Purkinje cell neurotransmission is needed for DBS to be effective (Fig. 6). In addition, one main location at which learning takes place within the cerebellum is at the parallel fiber-and-Purkinje cell synapse, whose activity is modulated by mossy fiber inputs. Mossy fibers relay information from the cortex and spinal cord to the cerebellum in order to refine behaviors. Simple spikes are an output measure of Purkinje cells that represent the action potentials generated by mossy fiber inputs. Here, we found that simple spike properties change in $Car8^{wdl}$ mice as motor function deteriorates (Fig. 6). When ataxia peaks in $Car8^{wdl}$ mice, Purkinje cells fire erratically on the whole due to increased pausing. Curiously, the high regularity we find in local Purkinje cell firing (CV2) disappears. Because local variation in simple spike firing is important for motor timing[59] and learning[75], these findings suggest that the regularity of Purkinje cell firing prior to stimulation may be important for predicting and sustaining responses. Although, we cannot discount the role that age and other circuit firing features play in DBS reception (Fig. 5). Fully elucidating the Purkinje cell properties that impact DBS efficacy (in the presence and absence of age-related changes) will be critical for further optimizing the technique.

No treatment modality to-date fully rescues ataxia, but the greatest benefits have resulted from improving cerebellar and muscle functions. Fryer et al. showed that combining exercise with a genetic rescue reduced ataxia in SCA1 mice[76]. In humans, pairing brain-computer interfaces with functional electrical stimulation of the muscle recovered voluntary motor control[77]. Here, we expanded on these results by showing that DBS

combined with motor training yields long-lasting improvements in ataxia. However, in order for DBS to be effective, certain deficits must exist prior to stimulation. This may include prolonged muscle contractions and highly regular trains of Purkinje cell firing. If prolonged muscle contractions are a prerequisite for DBS working, then cerebellar stimulation could also be beneficial for different forms of dystonia, a heterogeneous disorder in which specific abnormalities, such as widened EMG signals and muscle co-contractions, tend to occur as consistent features[43]. Preliminary studies in humans[78] and mice[26] indeed support the utility of cerebellar stimulation for dystonia. How effective therapeutic cerebellar DBS would be for neurodegenerative ataxias depends on which circuit properties change and how they might mediate the positive effects of stimulation because Purkinje cell loss alone does not block cerebellar DBS effectiveness[18]. Consistent with these data, cerebellar DBS in the *shaker* rat model of degenerative ataxia shows benefits[31]. In such degenerative models, Purkinje cell degeneration and eventual cell loss induces a plethora of circuit rearrangements, compensatory fiber growth, and death of surrounding cell types. Also, because Purkinje cell degeneration is often patterned[79], there is residual cellular function in surviving cells. Together, this may create a neural environment conducive to propagating the effects of stimulation. In such a scenario, it is also interesting to consider that DBS is effective when presented with multiple dysfunctional cell types and microcircuits upon which to act. Interestingly, although *Car8^wdl* mice do not have overt neurodegeneration, they do have altered cerebellar patterning[24]. While the distribution of pathological signatures could be more fully appreciated with additional *Car8^wdl* experiments and increasing the number of animals used here, our data strongly support a neurostimulation approach with high therapeutic potential. Our data additionally highlight the duplicity of cerebellar DBS effects: DBS corrects *Car8^wdl* behavioral deficits to support ongoing motor improvements during treatment, but it may also prompt learning mechanisms to sustain those improvements after the treatment regimen has been completed.

## Methods

**Animals**. All animal studies were carried out under an approved IACUC animal protocol according to the institutional guidelines at Baylor College of Medicine (BCM). All of the animals used in this study were maintained in our animal colony at BCM. The *Car8^wdl* mice (Stock #004625) and the C57BLKS/J control background strain were purchased from The Jackson Laboratory (Bar Harbor, ME). The *L7^Cre;Vgat^flox/flox* and the *Vgat^flox/flox* control littermates were genetically engineered, as previously described[63]. Note that in the literature, *L7* is also referred to as *Pcp2* (Purkinje cell protein 2) and *Vgat* is referred to as *Slc32a1* (solute carrier family 32 member 1). We bred the control and mutant mice using timed pregnancies, and we designated noon on the day a vaginal plug was detected as embryonic day (E) 0.5 and the day of birth as postnatal day (P) 0. We used a standard PCR genotyping protocol to differentiate the mutants from the controls, using the primer sequences listed in Supplementary Data 2[19,80,81]. Mice of both sexes and aged P30, P60–P120, and ≥P150 were studied. Biological sex did not impact the mouse behaviors that were studied (Supplementary Fig. 17) after surgery, nor did we observe significant behavioral differences between P60, P90, and P120 mice after surgery and by the end of training (Supplementary Fig. 17). However, the motor behaviors that were characteristic of the P30 and ≥P150 mutant mice were significantly different from that of P60–P120 *Car8^wdl* mice; therefore, three cohorts (P30, P60–P120, ≥P150) were generated and used in our studies (Supplementary Fig. 17). All of the mice had food and water ad libitum.

**Surgical procedures**. For all surgical techniques used in these studies, the mice were given preemptive analgesics (buprenorphine slow release, 1.0 mg/kg subcutaneous (SC), and meloxicam, 5.0 mg/kg SC) with continued applications provided as part of the post-operative procedures. Anesthesia was induced with 3% isoflurane gas and maintained during surgery at 2% isoflurane gas. All surgeries were performed on a stereotaxic platform (David Kopf Instruments, Tujenga, CA, USA) with sterile surgery techniques applied throughout the procedures. Immediately following surgery, all of the mice were placed in a warming chamber. Animals that underwent surgery for deep brain stimulation (± electromyography) were allowed to recover for at least 3 days before any additional experimental

analyses were performed. Mice used for WGA conjugated to Alexa 555 neuroanatomical tracing were perfused 24 h after the surgery.

**Deep brain stimulation**. Control and mutant mice underwent surgeries for DBS and EMG. Twisted bipolar tungsten electrodes with a width of 0.127 mm and a length of 3.5 mm were purchased from PlasticsOne for DBS. Two electrodes were spaced 2.6 mm apart to bilaterally target the interposed cerebellar nuclei and fixed together using Bondic, a UV-light activated bonding agent (Amazon). Next, P30, P60–P120, and ≥P150 mice (*n* ≥ 3 of each genotype) were deeply anesthetized with isoflurane so that the prepared DBS electrodes could be inserted into the cerebellum using the following stereotactic coordinates, as calculated from Paxinos and Franklin (2001): −6.4 mm (anterior–posterior), ±1.3 mm (medial–lateral), and −2.5 mm (dorsal–ventral)[82]. These coordinates were used to guide surgery at all of the ages studied because the size of the cerebellar nuclei remains unchanged from P23 to P300[83]. The DBS electrodes were secured to the head with C&B Metabond (Parkell, Inc., Edgewood, NY, USA, SKU: S380) and Teets 'Cold Cure' Dental Cement (A-M Systems, LLC, Carlsborg, WA, USA, Catalog #525000 and #526000). Mice were allowed to recover in their home cages for at least 3 days before starting the behavioral tests.

**Wheat-germ agglutinin (WGA)-Alexa 555 neuroanatomical tracing**. To prepare the tracer and equipment, a Drummond capillary tube (7″ #3-000-203-G/XL) was pulled using a micropipette puller (Narishige International Inc., East Meadow, NY), and backfilled with mineral oil[84]. Approximately 1 µl of 2% WGA-Alexa 555 (Cat. #W32464, Thermo Fisher Scientific, Waltham, MA) diluted in PBS was taken up into the tip of the pulled pipette. Control and *Car8^wdl* mutant mice (*n* = 3 per genotype) were then anesthetized and prepared for surgery as described above (see the section "Surgical procedures"). A unilateral craniotomy was made above the cerebellar interposed nucleus using stereotaxic coordinates (anterior–posterior: −6.24 mm, medial–lateral: +1.5 mm from Bregma) obtained from Paxinos and Franklin[82]. At a depth of −1.95 mm from the surface of the brain, 4 injections of 18.4 nL WGA-Alexa 555, each spaced 3 min apart, were administered. We used a bulk loading approach for neural tracing with WGA-Alexa 555 because of its reliability as an anterograde tracer when relatively large volumes are loaded. With these volumes, a substantial number of axons and terminals are robustly labeled and upon cutting the tissue, the signal is easily visualized due to the brightness of the signal. Note, however, that bulk loading is typically necessary since the uptake and transport of the tracer in individual cells cannot be controlled. The craniotomy was then filled with antibiotic and the skin was sealed with C&B Metabond (Parkell, Inc., Edgewood, NY, USA, SKU: S380), followed by Teets 'Cold Cure' Dental Cement (A-M Systems, LLC, Carlsborg, WA, USA, Catalog #525000 and #526000). Tissue was harvested 24 h later and prepared for imaging as described below (see the sections "Perfusion, basic histology, and tissue staining procedures" and "Image acquisition and quantification").

**Electromyography (EMG)**. To evaluate the effects of stimulation on hindlimb muscle activity, EMG and cerebellar DBS were combined. Under deep anesthesia, DBS electrodes were implanted to target the interposed cerebellar nuclei, as described above (see the section "Deep brain stimulation" under "Surgical procedures"), and in addition, two silver wires were inserted into the tibialis anterior (TA) and gastrocnemius (GC) muscles of the left hindlimb of control and mutant mice (*n* ≥ 3 of each genotype). A ground wire was implanted into the neck muscles and an EMG head mount was secured to the head with C&B Metabond (Parkell, Inc., Edgewood, NY, USA, SKU: S380) and Teets 'Cold Cure' Dental Cement (A-M Systems, LLC, Carlsborg, WA, USA, Catalog #525000 and #526000). Pre- and postoperative analgesics were provided, and the mice were given 3–4 days to recover before stimulation or the EMG recording sessions began. Muscle activity recordings were paired with either 0, 2, 13, or 130 Hz stimulation.

**In vivo electrophysiology**. P30 and ≥P150 *Car8^wdl* and control mice (*n* ≥ 3) were anesthetized with an intraperitoneal injection of a Ketamine (75 mg/kg)/Dexmedetomidine (0.5 mg/kg) cocktail. Anesthesia was maintained with 0.25% isoflurane. Anesthetized mice were placed into a stereotaxic frame and a craniotomy (~2 mm in diameter) was performed over the cerebellum. The craniotomy was drilled −6.4 mm from Bregma and 0 to +2 mm from the midline[82]. The electrodes were lowered from 0 to 2 mm, relative to the brain surface, to record Purkinje cell activity. Single-unit, extracellular recordings were then performed, using 5–8 MΩ tungsten electrodes and a motorized micromanipulator (MP-225; Sutter Instrument; also see the section "In vivo electrophysiology recordings and data analyses" under "Behavioral analyses").

## Behavioral analyses

**DBS and the accelerating rotarod**. To better understand the effects of DBS on animal behavior, DBS was paired with the accelerating rotarod to measure motor coordination and motor learning, with EMG to quantify muscle bursting, with the open field assay to track movement, and with footprinting and DigiGait analyses to assess gait. For the accelerating rotarod, there were three assessment periods: "Before ±DBS," "With or Without DBS," and "After ±DBS" following the surgical implantation of the DBS electrodes into the cerebellum. During each of these

assessment periods, the mice with surgically implanted DBS electrodes were attached to a system consisting of a Master8 pulse generator and an Iso-Flex stimulus isolator (AMPI, Jerusalem, Israel) and were allowed to acclimate for 5 min prior to being tested on the rotarod. The Master8-Iso-Flex system was not turned on for the "Before DBS" period (Days 1–4) so that no stimulation was delivered to the cerebellum. Mouse rotarod performance (i.e., latency to fall or time-to-1 rotation) was recorded for 4 consecutive days, with a baseline performance being established by Days 3 and 4. The rotarod (ENV-576M and ENV-571M, Med Associates, Inc., Vermont, USA) accelerates from 4 to 40 rpm in 5 min (setting 9) and was stopped at 300 s if the mice successfully stayed on for this duration of time. Mice rested for at least 10 min in between trials. The latency to fall for each mouse was recorded for a total of 4 trials each day, over a span of 18 days.

The Master8-Iso-Flex system was turned on for "With or Without DBS" (unless the mice were in the sham experimental group) and biphasic electrical pulses were subsequently delivered to the mouse cerebellum for 4 days (Days 8–11). The biphasic electrical pulses had the following parameters: a current of 30 μA, a duration of 60 μs, and a frequency of either 2, 13, 20, or 130 Hz. We chose 30 μA and 60 μs as the stimulation current and pulse width because these are the DBS parameters used in the clinic to treat various motor symptoms in humans[85–87], including dystonia and tremor, which are often ataxia co-morbidities. Shorter pulse lengths more broadly activate axons whereas smaller stimulation currents limit adverse side effects[87,88]. DBS was supplied during the 5-min acclimation period provided at the start of each of the 4 trials per day. Stimulation continued when the mice were on the rotarod and was stopped upon falling. When 13 and 130 Hz stimulation were "unpaired" from the rotarod, DBS was only provided for 5 min prior to the start of each trial and was not continued while the mice were placed on the rotarod. Each mouse that received DBS were stimulated for 20–40 min a day for 4 consecutive days. To determine whether DBS had a long-lasting impact on motor behavior, a 4-day "After ±DBS" period was implemented (Days 15–18). During this period, no additional stimulation was provided. Mice were connected to the Master8-Iso-Flex system (at that point, the device was left in the off position) and then given the chance to acclimate to the experimental setup for 5 min prior to the start of each of the 4 trials conducted per day.

Because control and Car8[wdl] mice exhibited variable starts on the rotarod after surgery, their motor performances when stimulated, or not stimulated, were normalized to their group baselines to visualize their individual improvements. We used the following equation: Normalized latency to fall = (average latency to fall "With or Without DBS," Days 8–11 or "After ±DBS," Days 15–18)/(average latency to fall baseline). Their baselines were determined by averaging their latency to fall from Days 3 and 4 of the "Before ±DBS" period because this is when their motor performances plateaued. The normalized latency to fall values for "Before ±DBS" and "With or Without DBS" (or "After ±DBS") were then plotted on a before-and-after graph. A two-way repeated-measures ANOVA with (post hoc analysis: Dunnett's multiple comparisons test) or without mixed-effects (post hoc analysis: Sidak's multiple comparisons test) was performed to calculate whether the latency to fall of stimulated and non-stimulated mice were significantly different ($p < 0.05$) over time.

The average latency to fall for the "With or Without DBS" and "After ±DBS" periods were also calculated so that the percent improvement could be determined ([% improvement = ("With or Without DBS" or "After ±DBS" average latency to fall − individual latency to fall baseline)/individual latency to fall baseline] * 100%). The percent improvements were first calculated for each individual mouse using a repeated-measures approach. Then, all of the percent improvements calculated within each experimental group were averaged. To assess the impact of inter-individual variability on overall motor improvements from DBS, analyses were performed in accordance with those described by Hendricks et al.[89]. P60–P120 animals stimulated at 13 or 20 Hz were separated by behavioral response. Car8[wdl] mice that had a percent improvement greater than that of Car8[wdl] shams plus the standard error of the mean (SEM) were considered "responders." Car8[wdl] mice that had a percent improvement less than or equal to that of Car8[wdl] shams plus the SEM were considered "non-responders." The latency to fall of the 13 and 20 Hz "responders" were plotted on X–Y graphs.

*DBS and EMG in freely moving mice.* EMG head mounts were connected to a pre-amplifier while the surgically implanted DBS electrodes on the same control and mutant mice were attached to a Master8-Iso-Flex system. TA and GC muscle activity was recorded from control and mutant mice ($n ≥ 3$ of each genotype and stimulation group) for a total of 40 min. There were 3 consecutive recording periods per animal: "Before ±DBS," "With or Without DBS," and "After ±DBS." Like the periods of the accelerating rotarod, no stimulation occurred during "Before" or "After ±DBS," but may have during the "With or Without DBS" period at either a frequency of 0, 2, 13, or 130 Hz. The DBS amplitude and duration remained at 30 μA and 60 μs, respectively. "Before ±DBS" and "After ±DBS" lasted for 10 min each and the "With or Without DBS" period lasted for 20 min. To determine whether prolonged cerebellar stimulation persistently alters TA activity, this 40-min paradigm was repeated for 3 consecutive days. For each recording, animals were freely moving in a clean, empty cage. Times when locomotion occurred, as defined by the alternating movement of the left and right hindlimbs, were documented. Data were collected and analyzed using Spike2 software (Versions 7.09 and 7.20). Locomotor periods where there were at least 3 consecutive TA bursts were analyzed using Burst Analysis in Spike2 (Version 7.20). The burst

analyses of multiple locomotor periods were averaged to achieve a single, representative value per animal per session. For each animal, at least one period lasted 30 s to ensure that TA activity was consistent between shorter and longer bouts of locomotion. The number of bursts over time and the mean burst length in the TA muscle were averaged among the animals and compared between-groups using an unpaired, two-tailed Student's *t*-test ($p < 0.05$). A two-way repeated-measures ANOVA ($p < 0.05$) followed by a Sidak's multiple comparisons test was used to determine if DBS significantly altered control or Car8[wdl] muscle properties within the same mouse over time. A three-way repeated-measures ANOVA ($p < 0.05$) followed by a Sidak's multiple comparisons test was used to determine if 13 Hz DBS significantly altered Car8[wdl] muscle properties relative to control mice, over time. A three-way repeated-measures ANOVA with mixed-effects ($p < 0.05$) followed by a Sidak's multiple comparisons test was used to determine if 13 Hz DBS significantly altered Car8[wdl] muscle properties relative to control mice, after 3 days.

*DBS and DigiGait analysis, footprint analysis, and the open field assay.* The open field assay, DigiGait, and footprint analyses revealed the effects of DBS on the general locomotion and gait of P30 and P60–P120 control and mutant mice. The Open Field Locomotion system (Omnitech Electronics, Inc., Columbus, OH, USA) was used to carry out the open field assay across a 4-week time period. During the first week, mice were acclimated to the Open Field room for 1.5 h, then placed in the Open Field apparatus, where their movement was recorded for 30 min. The movement time and number of locomotor episodes of each mouse were specifically analyzed due to our DBS targeting choice (Accuscan Fusion Software, Version 4.7). Because the interposed nucleus controls ongoing motor function, we postulated that its stimulation would alter the time or continuity of movement. White noise was provided in the background during the acclimation and the recording periods. Within 1–3 days of this initial open field recording (termed "Day 0"), DBS surgery was performed on these mice. Three-to-five days later, the mice that were previously tracked and operated on were reintroduced to the open field assay ("Before ±DBS", "Day 6"). Locomotor data corresponding to two additional time points ("With or Without DBS" and "After ±DBS") were collected 7 ("Day 13") and 14 days ("Day 20") after the "Before ±DBS" period, respectively. For each of these 4 time points, mice experienced the same 1.5-h acclimation period to the Open Field room and 30 min of tracking. The only two parameters that changed between the periods was that (1) the surgically implanted DBS electrodes on the mice were connected to a Master8-Iso-Flex hybrid system (STG4002, Multichannel Systems, Baden-Württemberg, Germany) "Before," "With or Without," and "After ±DBS" and (2) stimulation (2, 13, or 130 Hz) was provided to a subset of animals on Day 13. Sham mice received no stimulation but were still connected to a powered-off Master8-Iso-Flex hybrid system. In addition, mice that received DBS were stimulated for 2 h (i.e., the 1.5-h acclimation period plus the 30-min open field tracking period). DBS frequencies (2, 13, and 130 Hz) were programmed into the Master8-Iso-Flex hybrid system using MC Stimulus II (Version 3.4.4) software. For all of the behavioral tests, there were no obvious differences in performance between the sexes. Therefore, both sexes were analyzed, and an unpaired *t*-test ($p < 0.05$) was used to compare between-group differences in movement time when the frequency and period were held constant. A two-way repeated-measures ANOVA ($p < 0.05$) followed by a Dunnett's multiple comparisons test was used to compare same-group differences in activity levels before, during, and after surgery and ±DBS. An unpaired, two-tailed Student's *t*-test ($p < 0.05$) was used to compare between-group differences in P30 mice. A three-way repeated-measures ANOVA ($p < 0.05$) followed by a Sidak's multiple comparisons test was used to compare between-group differences in the activity levels of P30 control and mutant mice before, during, and after surgery and ±DBS.

Following the open field assay, mice were footprinted[90] and analyzed on the DigiGait (Mouse Specifics, Inc., Boston, MA, USA) during the same time points ("Day 0," "Day 6," "Day 13," and/or "Day 20"). For the footprint assay, the hindlimbs of the mice were coated with washable, non-toxic, acrylic blue (left), and red (right) paint (Crayola) while connected to the Master8-Iso-Flex hybrid system. Mice then walked across a 25 cm strip of white paper that was positioned inside a custom-made, open-top Plexiglas tunnel. Three sets of footprints were collected per animal per time point. The "stride," "sway," and "stance" were measured from the footprints in regions most reflective of natural gait (i.e., 3 straight, consecutive steps), averaged for each mouse, and compared via a three-way or a two-way repeated-measures ANOVA ($p < 0.05$) followed by a Sidak's multiple comparisons test. An unpaired, two-tailed Student's *t*-test ($p < 0.05$) was used if the frequency and periods were constant between the groups.

For DigiGait analysis, one 3-s video was recorded (DigiGait Imager Software (Version 16)), Mouse Specifics, Inc., Boston, MA, USA) for each mouse at each time point, as they walked at ~8 cm/s on an enclosed treadmill. Videos were then manually analyzed to compare the braking, propulsion, and swing of stimulated and non-stimulated control and mutant mice. The videos were first converted from AVI into MP4 files ("Free MP4 Converter," Version 6.2.23, http://www.anymp4.com/), then into JPEG images via the "Free Video to JPG Converter" software program (v. 5.0.101 build 201, https://www.dvdvideosoft.com/products/dvd/Free-Video-to-JPG-Converter.htm). Every frame was extracted from the MP4 videos. For each video, 4 frame numbers were recorded: the frame number where the paw fully-contacted the ground ("Frame #1"), the frame number where the heel of the paw was first lifted ("Frame #2"), the frame number right before the paw was completely lifted from the ground ("Frame #3"), and the frame number where the

paw re-contacts the ground to start a new step cycle ("Frame #4"). These numbers were recorded for each individual paw over the span of 16 consecutive steps. The time it took for the hindlimb paws to go from Frame #1 to Frame #2 represented the duration of "braking" (duration (s) = frame-time conversion factor ∗ ((Frame #2 − Frame #1) + 1)). Likewise, the time it took for the hindlimb paws to go from Frame #2 to Frame #3 represented the duration of the "propulsion" period (duration (s) = frame-time conversion factor ∗ ((Frame #3 − Frame #2) + 1)). The "swing" was calculated from the time it took for the hindlimb paws to go from Frame #3 to Frame #4 (duration (s) = frame-time conversion factor ∗ (Frame #4 − (Frame #3 ± 1))). The "Frame-Time Conversion Factor" was determined by a "Frame-to-Time" conversion calculator in ZapStudio (https://www.zapstudio.net/framecalc/). Each video was recorded at a frame rate of 157 frames/s and the total number of frames equaled the total number of JPEG images extracted. After receiving the total time of each video, the total time was divided by the total number of frames to get the final "Frame-Time Conversion Factor" (Frame-Time Conversion Factor = total video time (s)/total number of frames). Data were either displayed in a bar graph or an X–Y plot. In the bar graph, each data point represents an average between the two hindlimbs. Similarly, in the X–Y plots, each data point represents the average braking duration of the hindlimbs of $n \geq 3$ mice. A two- or three-way ANOVA ($p < 0.05$) with or without mixed-effects was used to determine if differences in forced gait were significantly changed within the same animal (post hoc analysis: Sidak's multiple comparisons test) and between-genotypes (post hoc analysis: Dunnett's multiple comparisons test) over time. An unpaired, two-tailed Student's t-test ($p < 0.05$) was used to determine whether the braking, propulsion, or the swing phases of locomotion in $Car8^{wdl}$ mutant mice differed from that of the control mice.

*In vivo electrophysiology recordings and data analyses.* Single-unit recordings were attained from at least 3 mice of each genotype with 5–8 MΩ tungsten electrodes (Thomas Recording, Germany) and digitized into Spike2 software (CED, England, Version 7.09). Purkinje cell traces of ≥100 s were analyzed with (Spike2 Version 7.20), MS Excel (Version 16.29.1, 19091700), and MATLAB (Version R2018b). We examined the simple spike properties of P30 and ≥P150 Purkinje cells over a pre-defined period of recordings, including the pause percent, coefficient of variation (CV), and the coefficient of variation of adjacent interspike intervals (ISIs; CV2). While the pause (ISI > 5 × mean ISI) percent measures how often spiking within a Purkinje cell is interrupted, the CV measures the variability in spiking (CV = standard deviation of ISIs/mean ISI of a given Purkinje cell). A high CV value indicates irregular firing. Irregular firing is defined by inconsistent intervals of time (ISI = seconds) between spikes. The CV2 measures the local firing regularity by calculating the variability of firing within a short period of two ISIs (CV2 = $2|ISI_{n+1} - ISI_n|/(ISI_{n+1} + ISI_n)$)[91]. The pause percent, CV, and CV2 were all reported as the mean ± standard error of the mean (SEM). Each data point represents the firing activity of an individual Purkinje cell. Statistical analyses were performed using unpaired, two-tailed Student's t-tests ($p < 0.05$).

**Perfusion, basic histology, and tissue staining procedures.** To check electrode targeting, survey for cerebellar damage, and analyze muscle histology, mice were anesthetized with 2,2,2-tribromoethanol (Avertin) and transcardially perfused first with 0.1 M PBS (pH 7.2) then with 4% paraformaldehyde (PFA) after surgery and behavioral assays were performed. Dissected brains and TA muscles from perfused mice were post-fixed in 4% PFA for at least 24 h. Non-stimulated and stimulated brains from control and mutant mice were transferred onto 2% agar for whole-mount imaging and then cryoprotected via sequential immersion in 18% sucrose in 0.1 M PBS (pH 7.2; 24 h) and 30% sucrose in 0.1 M PBS (pH 7.2; 24 h). Following cryoprotection, brain tissues were embedded and frozen in Tissue-TEK O.C.T. compound solution (Sakura Finetek USA), then coronally or sagittally cut on a cryostat into 40 μm sections and stored at 4 °C, free-floating in PBS (if tissue did not undergo WGA-Alexa 55 tracing) or mounted directly onto slides and covered with a coverslip (if the tissue underwent surgery for WGA-Alexa 55 tracing). TA muscles were not cryoprotected or frozen. Instead, TA muscles were embedded in paraffin. This first entailed dehydrating the muscles in overnight incubations of 70% ethanol, 95% ethanol, 100% ethanol, and chloroform. Then, the dehydrated tissue was transferred into plastic cassettes and immersed in two changes of paraffin (1 for 18 h and 1 for 2 h) in a 65 °C oven. The muscles were positioned for cross-sectioning, covered in hot paraffin, and then left overnight at room temperature until the paraffin solidified. Paraffin-embedded muscles were cut on a microtome into 12.5-μm sections, mounted onto slides using a warm water bath, and incubated overnight in a 65 °C oven.

Next, the cut tissue was stained using an established immunohistochemistry protocol (detailed below)[92], except for the tissue harvested from the WGA-Alexa 555 tracing experiment. After mounting the WGA-Alexa 555 tracer tissue onto slides, the sections were then imaged using the Zeiss AxioImage.M2. Brain tissue sections (within ±0.875 mm from the midline) that did not undergo WGA-Alexa 555 tracing were stained to assess cerebellar cell and motor circuit integrity. Brain tissue sections (±1.3 μm from the midline) and muscle sections were additionally stained to assess electrode targeting or overall tissue health. In short, brain tissue sections were either prepared for nuclear (Nissl and Hematoxylin & Eosin, H&E) or antibody staining whereas muscle tissue sections were only prepared for antibody staining. Nuclear staining confirmed electrode targeting and revealed the

cellular and structural integrity of the cerebellum, thalamus, and motor cortex. Antibody staining was used to assess surgical effects and muscle atrophy. Brain tissue designated for nuclear staining was mounted onto slides and then dried overnight. The slides containing the dried tissue sections were either submerged in hematoxylin for 5 min, lithium until the sections turned a deep blue, then eosin for ~1 min or Nissl solution for 5 min. Following eosin and Nissl staining, the brain tissue was dehydrated through sequential rinses in ethanol (70, 95, 100, 100, and 100%) and xylene twice. Slides were coverslipped using Richard-Allan cytoseal and left in the fume hood to dry overnight.

Brain and muscle tissue sections designated for antibody staining were prepared differently from that of tissue that underwent nuclear staining. The immunohistochemistry protocol followed was almost the same for both the brain and muscle sections prepared. The mounted slides containing the paraffin-embedded muscle sections underwent a series of washes to de-wax and rehydrate the tissue before staining began: 5 min in xylene, 2 min in 100% ethanol, 2 min in 95% ethanol, 2 min in 85% ethanol, and 2 min in 75% ethanol. Then, both brain and muscle tissue sections were incubated in 10% Normal Goat Serum (NGS, Sigma) blocking solution (0.1% Tween20 and 0.1 M PBS) for 2 h. After this incubation period, the 10% NGS blocking solution aliquots were replaced with new ones containing the desired primary antibodies. Tissue sections were incubated overnight (16–18 h).

A 1:500 dilution of an anti-rabbit laminin antibody (ab11575) was used with DAPI (Vectashield Anti-Fade Mounting Medium with DAPI #H-1200) to evaluate muscle health. The TA of stimulated and non-stimulated control and $Car8^{wdl}$ mutant mice were also stained with an anti-slow skeletal myosin heavy chain antibody (anti-mouse; ab11083) and an anti-fast myosin skeletal heavy chain antibody (anti-rabbit; ab91506), both at a 1:500 dilution, to determine myofibril composition. In addition, several antibodies were used to assess neuronal health. The calbindin D-28K monoclonal mouse antibody (Swant #300) and a NeuN anti-rabbit antibody (ABN78) were used to visualize Purkinje cells and granule cells at a 1:10,000 and 1:500 dilution, respectively. Anti-rabbit Iba1 (Fujifilm Wako 019-19741) at a 1:500 dilution marked microglia and anti-rabbit GFAP (DAKO Catalog #Z0334) at a 1:500 dilution marked astroglia to determine whether electrode implantation and/or stimulation resulted in extensive inflammation or glial scarring. To further survey for motor circuit damage, an anti-mouse neurofilament heavy chain (NFH) antibody and an anti-rabbit tyrosine hydroxylase (TH) antibody were used to visualize the cellular and structural integrity of the red nucleus and basal ganglia. The NFH (Covance Catalog #PCK-592P) and TH (Millipore Sigma AB152) antibodies were used at a 1:500 dilution after the brain tissue was incubated with 0.03% hydrogen peroxide ($H_2O_2$) in 0.1 M PBS for 5 min and washed with 0.1 M PBS in 3–5-min intervals. Brain tissue sections were incubated overnight (16–18 h). After the overnight incubation, brain tissues were washed 3–4 times with 0.1 M PBS for 5 min each. Brain tissues were then either incubated for 2 h with fluorescent Alexa 488-, 555-, or 647-immunoglobins (Invitrogen Molecular Probes Inc., Eugene, OR, USA #A-21202, #A-31572, and #A-31573) diluted 1:1500 in new aliquots of 10% NGS blocking solution, or with goat anti-mouse (Dako, P044701-2) and goat anti-rabbit (Dako, P044901-2) HRP-conjugated antibodies diluted 1:200 in new aliquots of 10% NGS blocking solution. Tissues were then additionally rinsed 3–4 times with 0.1 M PBS in 5-min increments. Afterward, the fluorescently stained brain tissue was mounted onto slides and coverslipped with FLUORO-GEL (with Tris Buffer) or DAPI (Vectashield Anti-Fade Mounting Medium with DAPI #H-1200). 3,3′-Diaminobenzidine (DAB) was added to the brain tissue that was previously incubated with the HRP-conjugated secondaries. After the brain tissue darkened, 0.1 M PBS was used to stop the colorimetric reaction. DAB-stained tissue was then mounted onto slides and left at room temperature overnight. The slides were transitioned from 50% ethanol, 70% ethanol, 2 sequences of 95% ethanol, 2 sequences of 100% ethanol, and xylene every 5 min until the tissue was dehydrated. Coverslips were secured to the slides using Richard-Allan Scientific™ Cytoseal™ XYL (Thermofisher Scientific, #8312-4). All of the stained tissues were imaged on the Zeiss AxioImage.M2, although the DAB-, Nissl-, and H&E-stained tissues were imaged under brightfield imaging settings.

**Image acquisition and quantification.** A Zeiss AxioZoom VI6 (Zeiss AxioVision Software, Release 4.8) was used to image the cerebella in whole-mount. A Zeiss AxioImage.M2 with Apotome equipped with Z-stack features (Zeiss Zen Pro Software, Version 2.0.0) was used to capture fluorescent images to survey for tissue damage after electrode implantation and stimulation as well as to assess muscle fiber-type composition. The Zeiss AxioImage.M2 (Zeiss Zen Pro Software, Version 2.0.0) was also used for brightfield microscopy. Various histological measures were quantified from the captured images, including myofibril composition, cerebellar molecular layer thickness, GFAP staining intensity, and Iba1 staining intensity. Slow, fast mixed, and unstained muscle fiber types were quantified as percentages from 3 consecutive TA tissue sections from three different animals, per genotype. Muscle fibers were considered "mixed" if the myofibril contained both slow and fast-twitch-related proteins. If a myofibril was not immunoreactive to the fast or the slow-twitch antibodies, then the myofibril was considered "unstained." The percentages from each tissue section were averaged per animal and then we conducted between-group comparisons using a three-way ANOVA ($p < 0.05$) followed by a Sidak's multiple comparisons test to analyze differences in the proportion of

slow- and fast-twitch muscle fibers in stimulated and non-stimulated control and *Car8wdl* mice. Calbindin-stained cerebellar tissue from stimulated and non-stimulated P60–P120 control and *Car8wdl* mice were used in ImageJ to measure molecular layer (ML) thickness. We define ML thickness as the distance from the Purkinje cell somata in the Purkinje cell layer (PCL) to the apical surface of the cerebellar cortex. We measured ML thickness in lobule IV/V of 0, 13, and 130-Hz stimulated mice ($n = 3$ controls, $n = 3$ mutants, per condition) due to its role in locomotion[25] and its connectivity to the primary motor cortex[93] and interposed nucleus[94]. Six measurements were taken from each animal, then averaged. ML thickness values were plotted in a bar graph, where each point represents the average ML thickness of an individual animal. A two-way ANOVA ($p < 0.05$) followed by a Tukey's multiple comparisons test was performed to determine whether ML thickness differed between genotypes and stimulation groups. GFAP and Iba1 staining intensities surrounding 6 electrode tracks from three different animals (per genotype, per condition) were quantified, using ImageJ software (Version 1.0). Each image was first duplicated, then the copy was converted into a 16-bit image. The threshold was adjusted on each 16-bit image so that the GFAP- or Iba1-positive areas were accurately highlighted. Using the "Image Calculator" processing function, ImageJ subtracted the established threshold from the original image, yielding a "Mean pixel intensity" value for GFAP or Iba1 staining. Plotted on bar graphs, each point represents the average GFAP or Iba1 staining intensity of an individual animal. A two-way ANOVA ($p < 0.05$) followed by a Sidak's multiple comparisons test was performed to determine whether the staining intensity of GFAP or Iba1 differed between genotypes and stimulation groups.

**Data and statistical summaries**. A summary of the data and statistical tests used throughout the study can be found in Supplementary Data 1. The two-tailed, unpaired Student's *t*-test ($p < 0.05$) was used whenever one measurement from mutant and control mice were being compared. The one-way ANOVA ($p < 0.05$) was performed when comparing one measurement from one genotype across different ages. The two-way repeated-measures ANOVA ($p < 0.05$) was used whenever one measurement from one genotype (*Car8wdl* or control) was being compared over time and across different stimulation paradigms. The three-way repeated-measures ANOVA ($p < 0.05$) was used whenever one measurement from both genotypes (*Car8wdl* and control) was being compared over time and across different stimulation paradigms. In order to limit false positives in our P60–P120 rotarod experiments, one two-way repeated-measures ANOVA was performed, which incorporated all of the possible stimulation paradigms (Supplementary Data 1; No Surgery, 0, 2, 13, 20, 130, unpaired 13, and unpaired 130 Hz). For instances when tissue was harvested from mice following the last day of ±stimulation on the rotarod (Day 11) or the EMG signals did not last until Day 3, a separate two-way or three-way ANOVA was conducted with mixed-effects, using the same baseline measurements established "Before" and/or during "±DBS" as the controls. The mixed-effects model uses the maximum likelihood method to estimate missing values. After the ANOVAs were performed, a Sidak's, Dunnett's, or Tukey's multiple comparisons test was used to compare the means of the different groups. A Sidak's multiple comparisons test was used when comparing a select set of means. A Dunnett's multiple comparisons test was used when comparing experimental means to a control mean. A Tukey's multiple comparisons test was used when comparing all means. Sample sizes were determined based on the statistical criteria for significance in observations rather than an a priori power analysis since the experiments conducted were dependent on the success and/or failure of DBS[95]. All attempts at replication were successful, unless the DBS electrodes were mistargeted. Therefore, no data were excluded from analysis unless the electrodes were mistargeted, as determined through post hoc tissue analyses, or the mice did not complete the rotarod paradigm through Day 11 (Supplementary Fig. 2b). All of the statistical tests were performed by Prism8 software (Version 8.4.3, 471). If significant, the data in Supplementary Data 1 are highlighted in purple.

**Reporting summary**. Further information on research design is available in the Nature Research Reporting Summary linked to this article.

## Data availability

All data and materials used are available in the main text or as supplementary materials. The authors will also provide data and materials upon request. Source data are provided with this paper.

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

## Acknowledgements

This work was supported by funds from Baylor College of Medicine (BCM) and Texas Children's Hospital, BCM IDDRC Grant U54HD083092 from the Eunice Kennedy Shriver National Institute of Child Health and Human Development (The IDDRC Neuropathology and Neurobehavior Sub-Cores were used for a portion of the anatomy and behavior experiments.), and by National Institutes of Neurological Disorders and Stroke (NINDS) R01NS089664 and R01NS100874 to R.V.S.

## Author contributions

L.N.M. and R.V.S. conceived and designed experiments. L.N.M., T.L., J.Z., M.E.V.D. H., J.B., J.J.W., and R.V.S. performed experiments. L.N.M., M.E.V.D.H., J.B., and R.V.S. analyzed data. L.N.M. and R.V.S. drafted the manuscript. All authors interpreted results, edited and approved the final manuscript.

## Competing interests

The authors declare no competing interests.
