## [Peer Review File · Nature Communications]

REVIEWER COMMENTS

Reviewer #1 (Remarks to the Author):

General comments:

This study aims to investigate whether pairing low frequency DBS in the interpositus deep cerebellar nuclei with exercise improves motor function in a non-degenerative ataxia mouse model. To accomplish this, authors employed 2 different behavioral tests (open field assay and rotarod) in different aged mice to assess changes in locomotion; evaluated sex as a variable; gait and locomotion was monitored with the DigiGait system for brake, stride and propulsion endpoints while dyed foot paw analyses revealed overall gait changes. Additionally, authors included transgenic mice depleted of GABA to surmise that PC dysfunction underlied DBS efficacy; in vivo electrophysiological EMG recordings was obtained in TA and GC muscle in behaving animals and single-units from PCs from anesthetized mice at different ages; lastly, post-mortem tissue was analyzed for changes in leg myofibril composition and brain tissue for inflammatory reactions and integrity of cerebello-thalamo-cortical circuitry.

The authors report that combining DBS at 13 and 20 Hz, with exercise, in Car8 ataxia mice improved motor function by shortening TA muscle burst length, altering myofibril composition and eliciting adaptive behavior affecting the step cycle. Interestingly, efficacy was sustained 7-days after DBS was turned off.

In general, the study exhibited good scientific rigor and has high potential for making a significant impact in the research and clinical domains. Specifically, these findings provided provocative and novel data for a promising translational neuromodulatory paradigm for spinocerebellar ataxia. It also posits that DBS of the interpositus improves on-line motor function by normalizing muscle function and off-line via behavioral adaptations for sustained improvements. With that in mind, there are lingering major and minor considerations that diminish the enthusiasm of this study in its current form. For instance, there needs to be more clarity and justification regarding some of the methodology, improved figure readability and more substantive discussion of the data. As a final note, more data would help to bolster some of the author's conclusions and claims. Addressing these points will elevate the scientific rigor and overall quality of the study.

Major comments:

1) A major tenet of this study is that interpositus DBS provides maximal efficacy when ataxic symptomology is less severe. This is supported by retrospective analyses of their data looking at responders and non-responders and trajectory of improvement at different age cohorts of Car8 mice. This notion can be further substantiated by including another ataxia mouse model, but stimulated when only mild symptomology presents (e.g., 5-8 week old B05 mice) before moderate to severe ataxic phenotypes develop with age-related neurodegeneration.

2) 20 Hz stimulation was used in the experimental design. Yet, this data is only shown in some instances and not in many others. Yet, including data showing 20 Hz DBS effects on TA-GC and TA burst lengths in figure 2, OFA data in figure 3 and rotarod data in figure 4, would strengthen the conclusion that a spectrum of beta frequencies improve ataxia by similar mechanisms.

3) Control animals improve with 130 Hz DBS with no efficacy at 13 and 20 Hz (fig. S3) whereas Car8 mice exhibit the opposite response. In addition, 13 Hz in control mice increased slow twitch muscles as it did in Car8 mice (fig. S8). Yet, control mice did not improve with 13 Hz (fig. S3) whereas car8 mice did (fig. 1). Please discuss why this is the case.

4) P30 mice perform much better at baseline (fig. S12) than P60-120 or P>150. Yet, it is not shown whether TA burst lengths differ in these different aged cohorts. Furthermore, efficacy in motor function persisted 7 days after DBS was turned off, but there is no data about whether TA bursts are still shortened during this time without DBS. It would also be informative knowing if TA-GC coupling changes with age, DBS and post-DBS. Figure 2C only provides control data at one cohort age.

5) Interpositus DBS in P30 mice have less variable efficacy than mice at P>100. Yet, how does authors control for differences in volume of tissue activated given differences in electrode-brain volume ratio but the same current amplitude? It may be that stimulation of dentate or fastigial deep cerebellar nuclei may have occurred in P30 vs. P100 mice. Some discussion is warranted.

6) There is no data showing PC ISI CVs for P30 and P120 Car8 mice. Yet, these animals have different baseline rotarod performance (fig. S12), which presumably are due to differences in irregular spiking phenotype. Similarly, figure 6 shows that Car8 mice at P>150 have similar irregular ISI CVs as P21. Therefore, the younger mice would presumably perform at similar impaired levels as the older mice. Is there data showing this, e.g., latency to fall data for P21 Car8 mice as in fig. S12?

7) Some animal groups comprise of 3 animals, which is probably underpowered if a power analysis was performed. This should be mentioned as a limitation of the study in the discussion.

8) Figure 3b suggests that 2 Hz and 130 Hz DBS decreased or increased movement, respectively. Yet, these stimulation frequencies do not affect TA burst length (fig. 2h). Therefore, can the authors discuss the disconnect between TA burst length and motor performance for these stimulation frequencies? In fact, TA bursts return to pathological lengths after DBS is turned off (figure 3c), but sustained improved motor function is still noted (fig. 4a,b). This contrasts with figure S6 with no change in TA burst lengths in controls at 130 Hz despite control animals with improved rotarod performance in figure S3a. Perhaps a discussion can clarify these points.

9) It's not clear how the statistical analyses were performed for pie-chart data, e.g., was a chi-square with multiple corrections performed? Furthermore, it's unclear why controls in figure 4e were not included with figure 4i data. The proportions appear similar in all 3 groups for stride cycles, especially given the sample sizes.

10) Figure 5h: no quantification of calcium imaging data is provided for Car8 mice. Also, no age-matched control data is presented. While it is acknowledged that this has already been reported elsewhere (White et al. 2016), this should either be just cited, or a more rigorous presentation of data provided for completeness. Similarly, some data is provided as supplemental figures. With that said, quantification of the data should still be included to support certain claims. For instance, it was stated that there is minimal glial scarring or inflammation in Car8 mice after DBS (fig. S4) or cerebellar cortex cell layers are similar between control and car8 mice (fig. S5b).

11) Using the L7-vGat floxed mice is an innovative approach to see if PC activity is necessary to

mediate improvements in motor function in Car8 mice with DBS. However, it is not clear how authors can separate effects of GABA depletion on motor function per se to surmise efficacy from DBS. In other words, depleting GABA is so disruptive to normal PC and cerebellar physiology that movement is already grossly affected irrespective of DBS as underscored in figure 6h (no surgery or 0 Hz groups). Please discuss.

Minor comments:

1) Manuscript title mentions that mice are “severely ataxic”. While this is symptomatically correct, it may not be the ideal word to use since it gives a false perception that this mouse model exhibits marked PC degeneration to recapitulate late stage ataxia, which it does not.

2) Page 10, line 224: “muscle firing disappear after stopping DBS (fig. 3e)”. Should be fig. 3c.

3) Figures 5a,b are unclear. First, is the data from OFA or rotarod experiment? In (a), there are numbers on top of the histograms. Are they meant to denote responders and non-responders and if so, how is this different from (b)? Also, it does not appear that all responders and non-responders are shown in (b), e.g., only 2 responders and 1 non-responder shown for P100 and only 1 responder shown for P60 etc., which does not align with n=5 sample size. Page 12 line 269 states that 66.7% responded. Yet, figure (a) has 3/5 but (b) has 2/3. The latter coincides with this value, but it’s not clear if 2/3 is shown correctly if n=5. Lastly, it isn’t clear why figs. 5a and 5b are necessary to show as separate sub-figures.

4) Total distance travelled was not included as an endpoint metric for the OFA data. Please explain why not given that this metrics may better represent altered stride cycles and gait pattern rather than time spent moving.

5) Page 13, line 280: “in contrast, when Car8 motor function is less deteriorated, P30 mutants did”. This is an awkward or incomplete sentence/thought.

6) Improvement from interpositus DBS in a mouse model of dystonia formed the scientific premise for this study. Yet, the authors also cite DBS in the dentate nucleus in the shaker rat model. Therefore, there should be an explanation why the dentate nucleus was not targeted for DBS in this study.

7) Page 15, line 318: “at beta-frequencies (10-30 Hz)” should be revised to underscore the specific beta frequencies used in the study (13, 20 Hz).

8) Although rationale is given for the different stimulation frequencies employed, how was the current amplitude of 30 uA selected, e.g., the optimization approach/decision tree?

9) The rationale to normalize rotarod data to the baseline is because Car8 mice exhibited variable start times. However, fig. S2a indicates otherwise. Please clarify.

10) Authors mention that cerebellar-thalamo-cortical circuitry remain intact supported by nissl stain images. However, injecting anterograde/retrograde tract tracing dye/viruses would be a better approach to support this conclusion.

11) In general, it was hard to discern changes for certain data points with symbols laid over top of each other with similar colors and symbols. This became more indiscernible when control data are shown alongside Car8 data, e.g., fig. 2f and fig. 4h. Is there another stylistic/graphical way to improve readability for this data?

12) Figure 4 caption states that “non-stimulated Car8 mice worsen over time”. Fig. 4g does not show worsening over time, but rather sustained impaired braking.

13) Figure 4h: asterisks appear to denote wrong group/symbol.

14) Figure S1: caption letters do not align with figure.

15) Figure S7: page 49, line 1050: “in control (e) and mutant (f)”. Letters do not align with figure/table.

16) Figure S8: Caption for (e) is worded incorrectly since statistics support findings, not pie charts. Also, there is no reference in the caption for (f).

Reviewer #2 (Remarks to the Author):

NCOMMS-20-12228 is an extensive report of novel DBS stimulation of the cerebellum, via the interposed nuclei, as a therapeutic strategy in a mouse model of severe cerebellar ataxia. The authors report that DBS stimulation when applied in conjunction with motor paradigms, can improve the motor phenotype. The major strength of this report is that the treatment promotes direct, on-line, motor improvement while also creating indirect, offline behavioral adaptation that facilitates long-term benefits.

The research is of clinical importance to degenerative ataxia patients and may be adaptable to a wide spectrum of diseases including the spinocerebellar ataxias, cerebellar ataxias and episodic ataxias as well as stroke and tumor cerebellar disease.

I recommend this manuscript for publication without any major revisions.

Minor revisions and suggestions are:

1) Lines 72-77: unclear wording. Ataxia is a symptom, not a disease. Correct wording such as "ataxia is an incurable motor disease".

2) Greater clarity on the choice of the interposed nuclei for DBS targeting is desirable. Were other sites targeted?

3) Rotarod data described in lines 132,133 needs units.

4) Lines 280-281 needs re-wording

5) Zoomed in close-up images of cerebellar PC IHC in the supplemental figures would be appreciated in order to better present the cytoarchitecture.

Reviewer #3 (Remarks to the Author):

The authors offer a very dense, very well-written manuscript, with multiple experiments and multiple controls aimed at examining the effects of cerebellar deep brain stimulation in a mouse model of ataxia. As the authors point out, ataxia and, in particular cerebellar ataxia, remains one of the disabling movement disorders without an adequate solution. In fact, one of the current partial exclusion criteria for offering patients deep brain stimulation for the management of movement disorders is the presence of ataxia. For example, when evaluating patients as potential candidates for the treatment of essential tremor, we tend to exclude those that presented with overlapping ataxia. Hence, this line of investigation is important not only to examine the opportunity to better characterize cerebellar pathways, but also to open new paths for a translational therapy that can be relevant to a patient population that today is under treated. The results of these experiments, including the main preclinical finding that corroborates the use of deep brain stimulation of the cerebellar nuclei for ataxia in the mouse model, are all very interesting. The putative mechanisms of action of cerebellar deep brain stimulation were thoroughly dissected and correlations were traced with several variables including age and viability of the Purkinje cell system. This reviewer has a few thoughts and concerns to be considered by the authors.

1. In the introduction and in the discussion the authors state that to date there are no paradigms that provide sustained benefits from deep brain stimulation after it is deactivated. This is not consistent with the practice and research of deep brain stimulation in movement disorders. While the carryover in patients with essential tremor can be minimal once electrical stimulation is stopped, carryover is present in patients with Parkinson's disease and dystonia. In fact, most studies evaluating the mechanisms underlying the effects of deep brain stimulation include washout periods in order to minimize the effects of carryover between phases of the research. This consideration is particularly significant in the management of dystonia. There is ample experience showing that dystonia does not immediately worsen once stimulation is discontinued as evidenced by patients that continue to do well for long periods of time after the pulse generator device dies. We suggest reconsidering this point around the novelty of carryover effects in deep brain stimulation.

2. Still in the introduction, in the lines 84 and 85, the authors state that initial attempts for treating motor deficits with cerebellar DBS were inconsistent. Are the authors referring to the work by Ross and Cooper in the 1980s? If so, we note that those were neuromodulation studies but not deep brain stimulation studies as the investigators utilized surface electrodes on the tentorial surface of the cerebellum.

3. Deep brain stimulation target. The authors elected to stimulate the interposed nucleus of the cerebellum, which makes perfect sense given the spinocerebellar mechanisms related to ataxia. However, it may be difficult to target the interposed nucleus in the mouse and to restrict the effects of stimulation to such small nuclei. One option would be to frame this simply as deep cerebellar stimulation, acknowledging that the effects may be resulting from stimulation of the medial and lateral nuclei, combined. Another option would be to estimate the extent of the current spread relative to lead location, utilizing charge density and electrical stimulation modeling.

4. The extensive examination of the effects of various frequencies of stimulation on behavior is probably the most impactful and interesting part of the study. The authors demonstrate the frequency-specificity the results, which further provides internal controls to validate the most robust findings associated with low-frequency stimulation. One finding that stood out but is not followed by significant commentary is regarding the effects of 130 Hz stimulation on control mice. It seems that

while low frequency stimulation improved motor control in ataxic animals, high-frequency stimulation enhanced motor control in normal controls. Is that the case? Please comment further on this very curious finding.

5. Another very relevant finding is that DBS normalizes muscle activity in the ataxia model during locomotion. The authors demonstrate the effects of DBS on overlapping co-contractions of the tibialis anterior and gastrocnemius. We note that this type of agonist/antagonist co-contraction is very characteristic of dystonia and question if the CAR8 mouse model is really a pure model of ataxia or if it could be a model of ataxia and dystonia. This question is relevant to the interpretation of the results as the authors may have found not only a therapeutic approach for ataxia but also a therapeutic approach that may be viable for some forms of dystonia.

6. The authors investigated the effects of DBS according to age and provided data that correlate younger age with stronger therapeutic response. A co-variable is duration of disease, which can also influence response, as noted by the authors. While it makes perfect sense that animals with younger age or less duration of disease may have better outcomes, this reviewer is concerned with this small sample size that led to this conclusion. Based on figure 5, it seems that there are only one to three animals in each age group. We could not find statistical analysis supporting these assessments and there is a chance that the finding is really not significant. It's probably wiser to defer this topic to a future study that will be dedicated at examining the effects of age and disease duration on motor outcomes with low frequency deep brain stimulation in the mouse model of ataxia.

7. Lines 300 through 316. The authors tested the effects of interposed nucleus DBS in mutant mice that lack GABA neurotransmission, learning that there was no benefit of DBS in that model. Is this finding sufficient to conclude that DBS needs to be initiated before the peak symptoms of ataxia? Can the findings from this second mouse model directly inform the mechanisms of DBS in the CAR8 mutant mice? Perhaps a safer way to examine the role of GABA on the mechanisms of DBS in the CAR8 model would be to selectively inhibit GABA in this same mouse model and evaluate if it alters the outcome. This reviewer is concerned about utilizing two different mouse models to infer the mechanisms of DBS in the first mouse model.

Response to reviewer comments for manuscript NCOMMS-20-12228A now entitled "**Therapeutic neuromodulation of the cerebellum rescues movement in ataxic mice**" performed by Lauren N. Miterko et al.

We would like to thank all three reviewers for their constructive comments on how to improve our manuscript. We have addressed the suggestions by making changes and additions to the text and/or figures according to the reviewer's comments. We have also added new data, as requested.

Below are our explanations for how we have altered the manuscript in this revised version. The Reviewer's comments are written in regular text, and our responses are written in bold.

Reviewer #1: This study aims to investigate whether pairing low frequency DBS in the interpositus deep cerebellar nuclei with exercise improves motor function in a non-degenerative ataxia mouse model. To accomplish this, authors employed 2 different behavioral tests (open field assay and rotarod) in different aged mice to assess changes in locomotion; evaluated sex as a variable; gait and locomotion was monitored with the DigiGait system for brake, stride and propulsion endpoints while dyed foot paw analyses revealed overall gait changes. Additionally, authors included transgenic mice depleted of GABA to surmise that PC dysfunction underlied DBS efficacy; in vivo electrophysiological EMG recordings was obtained in TA and GC muscle in behaving animals and single-units from PCs from anesthetized mice at different ages; lastly, post-mortem tissue was analyzed for changes in leg myofibril composition and brain tissue for inflammatory reactions and integrity of cerebello-thalamo-cortical circuitry.

The authors report that combining DBS at 13 and 20 Hz, with exercise, in Car8 ataxia mice improved motor function by shortening TA muscle burst length, altering myofibril composition and eliciting adaptive behavior affecting the step cycle. Interestingly, efficacy was sustained 7-days after DBS was turned off.

In general, the study exhibited good scientific rigor and has high potential for making a significant impact in the research and clinical domains. Specifically, these findings provided provocative and novel data for a promising translational neuromodulatory paradigm for spinocerebellar ataxia. It also posits that DBS of the interpositus improves on-line motor function by normalizing muscle function and off-line via behavioral adaptations for sustained improvements. With that in mind, there are lingering major and minor considerations that diminish the enthusiasm of this study in its current form. For instance, there needs to be more clarity and justification regarding some of the methodology, improved figure readability and more substantive discussion of the data. As a final note, more data would help to bolster some of the author's conclusions and claims. Addressing these points will elevate the scientific rigor and overall quality of the study.

We are pleased that the reviewer found our manuscript to have high potential for making a significant impact in the research and clinical domains. We have revised the text and figures to address their excellent suggestions below. We have also added substantial new data.

Major Point #1: A major tenet of this study is that interpositus DBS provides maximal efficacy when ataxic symptomology is less severe. This is supported by retrospective analyses of their data looking at responders and non-responders and trajectory of improvement at different age cohorts of *Car8* mice. This notion can be further substantiated by including another ataxia mouse model, but stimulated when only mild symptomology presents (e.g., 5-8 week old B05 mice) before moderate to severe ataxic phenotypes develop with age-related neurodegeneration.

We thank the reviewer for appreciating that DBS is most effective when ataxia symptoms are less severe and for their curiosity on how our findings extrapolate to mouse models of neurodegenerative ataxia. Our current manuscript includes two models of ataxia, the *Car8^{w^{dl}}* model and *L7^{Cre};Vgat^{flox/flox}* model. Between these two models we show that DBS is most effective in *Car8^{w^{dl}}* mice that have less severe symptoms (notably, we study *Car8^{w^{dl}}* mice both with milder and more severe phenotypes) and that at least some Purkinje cell signaling is necessary for the DBS to be effective. In our revised manuscript, we have better articulated how the two different mouse models mechanistically address how the severity of ataxia, and specifically the circuit deficits therein, impact the effectiveness of DBS outcomes.

The reviewer is correct that cerebellar DBS may work for neurodegenerative ataxias, but as we have shown here, some residual Purkinje cell function must be present. This idea was previously raised by Teixeira et al. (2015), who found that DBS improved ataxia in the presence of cerebellar degeneration. From these data, two critical questions emerged: (1) What Purkinje cell properties does DBS interact with? And, (2) How does the progressive loss of cells in neurodegenerative ataxias impact DBS efficacy? These questions are now addressed through comprehensive textual revisions that we have made to the discussion section of the paper (page 22, lines 500-511).

Along these lines, we fully agree with the reviewer that additional models of ataxia, such as the B05 (SCA1) model is very interesting to consider. In fact, we have done just that, in a collaboration with our colleague, Dr. Huda Zoghbi. We have used a SCA1 knock-in model to test whether DBS recues movement in a severe degenerative ataxia. Over the past 4 years, this project developed into an intense investigation of DBS in progressive ataxias. We hope that the reviewer agrees that this additional investigation is deserving of its own publication, not only because of the additional concepts that we tackle there, but also because that story is simply much too large to include in this current paper. In summary, because we fully appreciate the suggestion by the reviewer, we instead re-worked and expanded our discussion to give readers a more complete view of cerebellar DBS possibilities.

Major Point #2: 20 Hz stimulation was used in the experimental design. Yet, this data is only shown in some instances and not in many others. Yet, including data showing 20 Hz DBS effects on TA-GC and TA burst lengths in figure 2, OFA data in figure 3 and rotarod data in figure 4, would strengthen the conclusion that a spectrum of beta frequencies improve ataxia by similar mechanisms.

Apologies for the confusion on how we presented our data. Our intent was to focus on the 13 Hz frequency, but to have some analysis on the 20 Hz frequency as a way of supporting the conclusion that multiple low frequencies are effective. DBS in either beta-frequency we tested

(13 and 20 Hz) improved ataxia. Although we did not analyze how 20 Hz DBS alters TA burst length or locomotion, we postulated that the results would be comparable to what we found with 13 Hz, given the equivalence of effects previously reported for these frequencies on behavior (Anderson et al., 2019) and circuit activity (Baker et al., 2010). For instance, both 10 and 20 Hz comparably improve the fall rate of rodents (Anderson et al., 2019) and induce motor-evoked potentials (Baker et al., 2010). These data raise the possibility that 20 Hz cerebellar DBS maintains the motor gains that *Car8^{w^{dl}}* mice achieve with 13 Hz. We tested this and found that 20 Hz cerebellar DBS similarly improves the rotarod performances of *Car8^{w^{dl}}* mice (Fig. S14). Since 20 Hz cerebellar stimulation may enhance cortical excitability over that of 13 Hz (Baker et al., 2010), we also calculated the response rate. We found that 20 Hz cerebellar DBS did not enhance the response rate and conclude that 13 and 20 Hz are likely functioning through a similar mechanism. To address fully the reviewer’s comment, we added the above citations to our manuscript (e.g., page 13, lines 298-301) as well as expanded our discussion (see page 20, paragraph starting with line 438). Furthermore, we now present our data more systematically. This meant that we present our 20 Hz data later in our manuscript, along with Fig. 5, where we felt that it more appropriately fits (page 13, “Ataxia severity contributes to the variability of DBS outcomes”).

Major Point #3: Control animals improve with 130 Hz DBS with no efficacy at 13 and 20 Hz (fig. S3) whereas *Car8* mice exhibit the opposite response. In addition, 13 Hz in control mice increased slow twitch muscles as it did in *Car8* mice (fig. S8). Yet, control mice did not improve with 13 Hz (fig. S3) whereas *car8* mice did (fig. 1). Please discuss why this is the case.

The differential effects that DBS has on control behavior may be explained by their starting points. For example, in our rotarod experiments, after the addition of more animals, only control mice that started off significantly worse (0 and 130 Hz) or with highly variable performances (2 Hz) improved with DBS (No surgery, $n=10$, 267.313 ± 5.601 s; 0 Hz, $n=15$, 194.358 ± 15.628 s; 2 Hz, $n=6$, 200.757 ± 18.544 s; 130 Hz, $n=11$, 187.591 ± 15.147 s; No/0 Hz, $p=0.0076$; No/2 Hz, $p=0.1410$; No/130 Hz, $p=0.0058$). These improvements equalized, rather than enhanced, control motor function (Fig. S3). These results suggest that control mice experience a ceiling effect and that their motor improvements depend on an initial presence of behavioral deficits—e.g., due to implant lesions. Our muscle composition findings (Fig. S9) support this conclusion by revealing a similar upregulation of the slow-twitch fiber protein in the TA of control and mutant mice. This is now included in our revised discussion on page 20, starting at line 438.

Because our histology data primarily shows how losing *Car8* does not interfere with how the muscle responds to cerebellar stimulation, we realized that its previous placement in our manuscript (which was alongside of our behavioral data) was confusing, especially when trying to interpret how long-lasting changes to muscle composition support long-lasting changes to motor behavior. Therefore, we now find it more fitting to discuss our muscle histology data when we consider whether *Car8^{w^{dl}}* motor circuitry is intact after surgery. Please refer to page 9, lines 193-201. We also use these data to support the hypothesis of a potential ceiling effect, as discussed earlier (page 20, lines 447-458).

Major Point #4: P30 mice perform much better at baseline (fig. S12) than P60-120 or P>150. Yet, it is not shown whether TA burst lengths differ in these different aged cohorts. Furthermore, efficacy in motor function persisted 7 days after DBS was turned off, but there is no data about whether TA bursts are still shortened during this time without DBS. It would also be informative knowing if TA-GC coupling changes with age, DBS and post-DBS. Figure 2C only provides control data at one cohort age.

We agree that it would be valuable to know how TA pathology differs with age and with treatment. We cite our previous work in Miterko et al. 2019 (page 10, line 211-217), in which we had already addressed TA burst properties in young (P21) control and *Car8^{w^{dl}}* mice. To investigate how TA burst properties change with aging, we now include EMG data from \geq P150 control and *Car8^{w^{dl}}* mice in a new supplemental figure (Fig. S16; page 14, lines 322-323). Together, our EMG analyses revealed that that TA burst lengths are comparable across ages in control and *Car8^{w^{dl}}* mice.

With regards to determining whether TA burst length differs 7 days after 13 Hz DBS, we tried performing this experiment, but unfortunately the EMG signal could not be reliably maintained for that long. We suspect that a combination of wire integrity and wire drift over time negatively impacts the signal quality. However, we did include in our manuscript new data that we confidently acquired from Day 3 (Fig. 3, S12; page 11, lines 252-253), which shows that TA pathology is further suppressed after repeatedly stimulating the cerebellum of *Car8^{w^{dl}}* mice at 13 Hz, as done with the rotarod. Therefore, although we could not follow the signal out to 7 days, we are confident that the effects on the muscle are not transient.

Major Point #5: Interpositus DBS in P30 mice have less variable efficacy than mice at P>100. Yet, how does authors control for differences in volume of tissue activated given differences in electrode-brain volume ratio but the same current amplitude? It may be that stimulation of dentate or fastigial deep cerebellar nuclei may have occurred in P30 vs. P100 mice. Some discussion is warranted.

We do not expect that there is a significant change in tissue volume that is activated in P30 mice versus \geq P100 mice because the size of cerebellar nuclei as a whole is not changed between these ages (Heckroth, 1994). We have now added this citation to our methods section (page 25, lines 572-573). In addition though, to fully account for the possibility that the DBS signal could, in theory, spread to other cerebellar nuclei and to be balanced in our wording, we replaced some of our strongly-worded references to “interposed DBS” to “cerebellar DBS.” In the discussion, we also explicitly acknowledged the possibility that the applied electrical stimulus could extend medially to the fastigial or laterally to the dentate nuclei (page 18, lines 409-411).

Major Point #6: There is no data showing PC ISI CVs for P30 and P120 *Car8* mice. Yet, these animals have different baseline rotarod performance (fig. S12), which presumably are due to differences in irregular spiking phenotype. Similarly, figure 6 shows that *Car8* mice at P>150 have similar irregular ISI CVs as P21. Therefore, the younger mice would presumably perform at similar impaired levels as the older mice. Is there data showing this, e.g., latency to fall data for P21 *Car8* mice as in fig. S12?

We agree that the conclusions of our manuscript would be strengthened if we include data on the PC ISI CVs for each aged cohort that was stimulated (P30, P60-P120, and \geq P150). Therefore, we now incorporated electrophysiology data from P30 *Car8^{w^{dl}}* mice in our revised manuscript (see the revised Fig. 6 and results under the subheading, “Purkinje cell firing properties change with motor deterioration in *Car8^{w^{dl}}* mice”). Our original draft already included electrophysiology from \geq P150 mice (Fig. 6) and discussed the results from P60-P90 mice (through reference to our previously published data in White et al., 2016, and Miterko et al., 2019).

To confirm that irregular ISI CVs correlate to impaired motor function in *Car8^{w^{dl}}* mice, we have now recorded the latency to fall for P21 mice, as per the reviewer’s recommendation. We found that although P21 *Car8^{w^{dl}}* mice are ataxic (Fig. S2), they outperform mutants in all of the other aged cohorts (Table S1). These data suggested that it is not the absolute CV value that matters, but rather, the extent of deviation from controls. We note that the regularity of local Purkinje cell firing (CV2) may also be important. Therefore, we revised the text under “Purkinje cell firing properties change with motor deterioration in *Car8^{w^{dl}}* mice ” to expand and clarify these points with a more balanced and complete discussion (page 15, starting at line 328).

Major Point #7: Some animal groups comprise of 3 animals, which is probably underpowered if a power analysis was performed. This should be mentioned as a limitation of the study in the discussion.

We have addressed this comment with two changes. First, we have increased the number of animals in several of our experiments. The new data with increased numbers reflect a higher degree of power. Second, to account for instances where the number of animals did not change, we have now added a sentence to our discussion to acknowledge this limitation, as recommended (page 23, lines 511-513).

Major Point #8: Figure 3b suggests that 2 Hz and 130 Hz DBS decreased or increased movement, respectively. Yet, these stimulation frequencies do not affect TA burst length (fig. 2h). Therefore, can the authors discuss the disconnect between TA burst length and motor performance for these stimulation frequencies? In fact, TA bursts return to pathological lengths after DBS is turned off (figure 3c), but sustained improved motor function is still noted (fig. 4a,b). This contrasts with figure S6 with no change in TA burst lengths in controls at 130 Hz despite control animals with improved rotarod performance in figure S3a. Perhaps a discussion can clarify these points.

Thank you for pointing this out. We agree that our initial description appeared to raise a disconnect between the EMG and rotarod data. We apologize that we did not originally clarify what changes to TA burst length means in our data. TA burst length does not relate to the speed or the total movement of the animals, but rather to motor timing and precision (Smith et al., 1977; Hadzipasic et al., 2016). We describe this on page 10 (lines 211-217) and now also emphasize this point in our revised discussion (page 19, paragraph starting at line 423).

With regards to why control mice improve on the rotarod, we address this in our response to Major Point #3. While 130 Hz DBS reduces TA burst length in 75% of stimulated controls (Fig. S6; $n=3/4$), it is likely a combination of factors contributing to rotarod improvements. For example, DBS could affect the function of other muscles that are important for locomotion. We now account for this possibility in our revised discussion (page 19, paragraph starting at line 423).

Major Point #9: It's not clear how the statistical analyses were performed for pie-chart data, e.g., was a chi-square with multiple corrections performed? Furthermore, it's unclear why controls in figure 4e were not included with figure 4i data. The proportions appear similar in all 3 groups for stride cycles, especially given the sample sizes.

To accommodate the new data, we no longer plot our DigiGait data in pie charts (please see the revised Fig. 4). For our DigiGait data, we used either an unpaired Student's t-test or a Three-Way Repeated Measures ANOVA. All of the statistical tests that we performed are described in our methods and in Table S1.

Major Point #10: Figure 5h: no quantification of calcium imaging data is provided for Car8 mice. Also, no age-matched control data is presented. While it is acknowledged that this has already been reported elsewhere (White et al. 2016), this should either be just cited, or a more rigorous presentation of data provided for completeness. Similarly, some data is provided as supplemental figures. With that said, quantification of the data should still be included to support certain claims. For instance, it was stated that there is minimal glial scarring or inflammation in Car8 mice after DBS (fig. S4) or cerebellar cortex cell layers are similar between control and car8 mice (fig. S5b).

To accommodate Major Points #6 and #11 and other revisions, we completely revised Figs. 5 and 6, as well as their text. As a result, Fig. 5h no longer presents data about tyrosine hydroxylase (TH) expression in Purkinje cells, which typically arises in the cerebellum after cellular changes in calcium handling. We found it sufficient to just talk about our previously published data (White et al., 2016), as appropriately indicated by the reviewer, and also cite the relevant literature as part of this discussion.

We now quantified our histology data presented in Fig. S4. For simplicity, we divided the original Fig. S4 into Figs. S4 and S8. In Fig. S4, we quantified Purkinje cell molecular layer thickness. In Fig. S7, we quantified glial scarring and microglial activation. The data are reported and described in the text under the subheading "Surgical lesions caused by DBS electrodes do not compromise motor circuit integrity." We have updated the methods section under the subheading, "Image Acquisition and Quantification," to include a description on how these data were quantified.

Major Point #11: Using the L7-vGat floxed mice is an innovative approach to see if PC activity is necessary to mediate improvements in motor function in Car8 mice with DBS. However, it is not clear how authors can separate effects of GABA depletion on motor function per se to surmise efficacy from DBS. In other words, depleting GABA is so disruptive to normal PC and cerebellar physiology that movement is already grossly affected irrespective of DBS as underscored in figure 6h (no surgery or 0 Hz groups). Please discuss.

This is a good point. First, for clarity, we now separately discuss our electrophysiology data and our $L7^{Cre};Vgat^{flox/flox}$ data under the subheadings, “Purkinje cell firing properties change with motor deterioration in $Car8^{wdl}$ mice” and “Cerebellar DBS relies on Purkinje cell neurotransmission,” respectively. Second, in the latter section (pages 16-17), we address the impact of GABA depletion on rodent motor function, specifically by discussing how our results are likely not specific to DBS (lines 372-382). For instance, we found in a previous study that administering a tremorgenic drug to $L7^{Cre};Vgat^{flox/flox}$ mice did not result in a behavioral change (Brown et al., 2020). Thus, the value of this model in which GABAergic neurotransmission is blocked (for clarity, GABA is still present, it is just that it is blocked from uptake into presynaptic vesicles) is the insight one gains from its circuit dysfunction, noting however that loss of GABA signaling would most likely cause molecular changes. We have now made this point clear in the text. In summary, this genetically-based circuit analysis approach allowed us to test whether a closed-loop cerebellar circuit is needed to propagate or rescue motor dysfunction in ataxic mice.

Minor Point #1: Manuscript title mentions that mice are “severely ataxic”. While this is symptomatically correct, it may not be the ideal word to use since it gives a false perception that this mouse model exhibits marked PC degeneration to recapitulate late stage ataxia, which it does not.

Agreed. We revised our title to now read, “Therapeutic neuromodulation of the cerebellum rescues movement in ataxic mice.”

Minor Point #2: Page 10, line 224: “muscle firing disappear after stopping DBS (fig. 3e)”. Should be fig. 3c.

Thank you for catching this typo. We have corrected the citation to Fig. 3c.

Minor Point #3: Figures 5a,b are unclear. First, is the data from OFA or rotarod experiment? In (a), there are numbers on top of the histograms. Are they meant to denote responders and non-responders and if so, how is this different from (b)? Also, it does not appear that all responders and non-responders are shown in (b), e.g., only 2 responders and 1 non-responder shown for P100 and only 1 responder shown for P60 etc., which does not align with n=5 sample size. Page 12 line 269 states that 66.7% responded. Yet, figure (a) has 3/5 but (b) has 2/3. The latter coincides with this value, but it’s not clear if 2/3 is shown correctly if n=5. Lastly, it isn’t clear why figs. 5a and 5b are necessary to show as separate sub-figures.

We have extensively revised Fig. 5 at reviewer recommendations, to more clearly convey our conclusions and to accommodate our increased number of mice (see response to Major Points #2 and #7). Therefore, you will find that we eliminated the bar graphs showing which $Car8^{wdl}$ mice responded to 13 Hz stimulation (previous Fig. 5a-b). Now, we include in Fig. 5a a breakdown of which mutants responded to 13 Hz DBS, using a pie chart. We included a rotarod schematic in Fig. 5a as well as labeled Fig. 5b with “Rotarod + 13 Hz” to make it clear which behavioral paradigm was studied. For simplicity, we moved the “Rotarod + 20 Hz” line graphs of the responders to a supplemental figure (Fig. S14) and only show the

“Rotarod + 13 Hz” data in the main figure (Fig. 5). Please see our response to Major Point #2 for a more in-depth discussion of the 20 Hz rotarod data.

Instead of a timeline, Fig. 5c-d now addresses how we found the most robust motor improvements in *Car8^{w^{dl}}* mice whose motor function was the least deteriorated by using the same before-after bar graphs that we used in previous figures (e.g., Fig. 1). We removed the example footprint traces and summary table from the revised Fig. 5 to streamline our message, which is that ataxia severity impacts DBS responsiveness.

Minor Point #4: Total distance travelled was not included as an endpoint metric for the OFA data. Please explain why not given that this metrics may better represent altered stride cycles and gait pattern rather than time spent moving.

We chose to measure the movement times of stimulated and non-stimulated mice because of our DBS targeting. The interposed nucleus controls ongoing motor function; therefore, we speculated that stimulating the interposed nucleus would alter the time or continuity (i.e., locomotor episodes) of movement. We added this justification to our methods section (page 32, lines 720-722).

Minor Point #5: Page 13, line 280: “in contrast, when Car8 motor function is less deteriorated, P30 mutants did”. This is an awkward or incomplete sentence/thought.

We have completely revised this section of the text to accommodate Major Points #1 and #6. Therefore, this line no longer exists in our revised manuscript. For your reference, the revised text can be found under the subheading “Purkinje cell firing properties change with motor deterioration in *Car8^{w^{dl}}* mice.”

Minor Point #6: Improvement from interpositus DBS in a mouse model of dystonia formed the scientific premise for this study. Yet, the authors also cite DBS in the dentate nucleus in the shaker rat model. Therefore, there should be an explanation why the dentate nucleus was not targeted for DBS in this study.

We expanded our explanation and rationale for choosing the interposed nucleus on page 5. This included referencing its specific role in locomotor adaptation (Darmohray et al., 2019), its composition of cerebellospinal projecting neurons (Sathyamurthy et al., 2020), and innervation by feedback collaterals (Beitzel et al., 2017). These features, in addition to its generally powerful influence on ongoing motor function, inspired our previous and recent studies to target the interposed cerebellar nucleus with DBS. We reported that interposed stimulation is very effective in reducing both dystonia and tremor in mice (White and Sillitoe, 2017; Brown et al., 2020). We have also revised our rationale for how a recent excellent study (Anderson et al., 2019) motivated our current experimental design by clarifying how stimulating the interposed nucleus, instead of the dentate nucleus, while the *Car8^{w^{dl}}* mice were engaged on the accelerating rotarod may boost DBS efficacy. We argue that the interposed is an ideal target for our purpose because it is specifically involved in sensorimotor processing during ongoing motion and locomotor adaptation, as we have now cited in the text.

Minor Point #7: Page 15, line 318: “at beta-frequencies (10-30 Hz)” should be revised to underscore the specific beta frequencies used in the study (13, 20 Hz).

Thank you. We no longer group together the beta-frequencies in our conclusions and instead specify the specific frequency we used in each presented experiment.

Minor Point #8: Although rationale is given for the different stimulation frequencies employed, how was the current amplitude of 30 uA selected, e.g., the optimization approach/decision tree?

Please see page 29 (lines 647-650) or below, where we added two sentences that clarify our decision for using an amplitude of 30 μ A and a pulse duration of 60 μ s.

“We chose 30 μ A and 60 μ s as the stimulation current and pulse width because these are the DBS parameters used in the clinic to treat various motor symptoms in human⁸⁵⁻⁸⁷, including dystonia and tremor, which are often ataxia co-morbidities. Shorter pulse lengths more broadly activate axons whereas smaller stimulation currents limit adverse side effects^{87,88}.”

Minor Point #9: The rationale to normalize rotarod data to the baseline is because Car8 mice exhibited variable start times. However, fig. S2a indicates otherwise. Please clarify.

Thank you for catching this. Our sentence in the methods is missing “after surgery.” Please see page 29 (line 662) and below for the revised sentence.

“Because control and Car8^{w^{dl}} mice exhibited variable starts on the rotarod after surgery, their motor performances when stimulated, or not stimulated, were normalized to their own baseline values to visualize their improvements.”

Minor Point #10: Authors mention that cerebellar-thalamo-cortical circuitry remain intact supported by nissl stain images. However, injecting anterograde/retrograde tract tracing dye/viruses would be a better approach to support this conclusion.

We agree with this comment and performed anterograde tracing in control and mutant mice, using Wheat Germ Agglutinin (WGA) Alexa-555. Results from this experiment confirm that projections from the interposed cerebellar nuclei of Car8^{w^{dl}} mice (n=3) are intact and that these projections to the thalamus and red nucleus are comparable to the projections we find in age-matched controls (n=3). We included images from this tracing experiment in Fig. S7.

Minor Point #11: In general, it was hard to discern changes for certain data points with symbols laid over top of each other with similar colors and symbols. This became more indiscernible when control data are shown alongside Car8 data, e.g., fig. 2f and fig. 4h. Is there another stylistic/graphical way to improve readability for this data?

To address this concern, we have converted the lines from all of the before-after plots into bars. Individual data points overlay each bar. Now every individual data point represents

one animal and every bar represents the sample mean \pm SEM. Please refer to Fig. 1, 2, 3, 4, 5, 6, S1, S3, S6, S10, S11, S13, S14, S15, and S16.

Minor Point #12: Figure 4 caption states that “non-stimulated Car8 mice worsen over time”. Fig. 4g does not show worsening over time, but rather sustained impaired braking.

Thank you for bringing this to our attention. We have revised the legend text accordingly. It now reads: “The braking deficits of non-stimulated *Car8^{w^{dl}}* mice are sustained over time.”

Minor Point #13: Figure 4h: asterisks appear to denote wrong group/symbol.

We have revised the appearance of our graphs to improve their readability (see our response to Minor Point #11).

Minor Point #14: Figure S1: caption letters do not align with figure.

We revised our legend text to better align with Fig. S1.

Minor Point #15: Figure S7: page 49, line 1050: “in control (e) and mutant (f)”. Letters do not align with figure/table.

Due to our revisions, Fig. S7 is now Fig. S11. We revised the letters in the legend parentheses to f and g, to properly match the new figure panels. We also updated the graphs, as was requested in Minor Point #11.

Minor Point #16: Figure S8: Caption for (e) is worded incorrectly since statistics support findings, not pie charts. Also, there is no reference in the caption for (f).

Due to our revisions, Fig. S8 is now Fig. S9. Still, we addressed this comment and revised the legend to more accurately reflect what is being shown in each panel. Please see below.

“Fig. S9. Cerebellar DBS increases the expression of slow-twitch muscle proteins. (a) Schematic of the intact mouse TA muscle and its cross-section. Magenta represents fast-twitch myofibrils while green represents slow-twitch myofibrils. In the cross-section, staining for fast and slow-twitch muscle proteins reveals fibers that are either slow (green), fast (magenta), putatively fast (unstained), or mixed (green and magenta). (b-d) 13 Hz cerebellar DBS increases slow-twitch protein expression in both the control and mutant TA muscle, without overtly affecting fast-twitch protein expression. (e-f) The percent of slow-twitch fibers increases after 13 Hz stimulation in control and *Car8^{w^{dl}}* mutant mice. The scale bar represents 50 μ m (b-d).”

Reviewer #2: NCOMMS-20-12228 is an extensive report of novel DBS stimulation of the cerebellum, via the interposed nuclei, as a therapeutic strategy in a mouse model of severe cerebellar ataxia. The authors report that DBS stimulation when applied in conjunction with motor paradigms, can improve the motor phenotype. The major strength of this report is that the treatment

promotes direct, on-line, motor improvement while also creating indirect, offline behavioral adaptation that facilitates long-term benefits.

The research is of clinical importance to degenerative ataxia patients and may be adaptable to a wide spectrum of diseases including the spinocerebellar ataxias, cerebellar ataxias and episodic ataxias as well as stroke and tumor cerebellar disease.

I recommend this manuscript for publication without any major revisions.

We are pleased that the reviewer found our manuscript to have far-reaching clinical implications for disease. We addressed the comments below to improve our manuscript.

Minor Point #1: Lines 72-77: unclear wording. Ataxia is a symptom, not a disease. Correct wording such as "ataxia is an incurable motor disease".

Thank you. In our revised manuscript, we no longer refer to ataxia as a disease. Therefore, sentences that had referenced ataxia as a “disease” were deleted or re-worded. For example, please see our revisions to the abstract or line 64 of the introduction.

Minor Point #2: Greater clarity on the choice of the interposed nuclei for DBS targeting is desirable. Were other sites targeted?

This is an excellent point. We addressed this concern in our revised manuscript. Please see our response above, to Major Point #6 from Reviewer #1.

Minor Point #3: Rotarod data described in lines 132,133 needs units.

We reported the rotarod data from Fig. 1 as percent improvements. Therefore, the “%” was included as the unit for the results under “Beta-frequency DBS into the cerebellum improves *Car8^{w^{dl}}* motor behavior.”

Minor Point #4: Lines 280-281 needs re-wording.

Thank you. We have revised this section according to comments received from all three reviewers. Therefore, this sentence (now page 14, lines 323-325) was replaced with: “*Together, these data indicate that a positive behavioral outcome to cerebellar DBS partially depends on the severity of ataxia at the time when treatment is initiated.*”

Minor Point #5: Zoomed in close-up images of cerebellar PC IHC in the supplemental figures would be appreciated in order to better present the cytoarchitecture.

We have now included zoomed-in images of cerebellar Purkinje cells from stimulated and non-stimulated control and *Car8^{w^{dl}}* mutant mice. Please refer to our revised Fig. S4.

Reviewer #3: The authors offer a very dense, very well-written manuscript, with multiple experiments and multiple controls aimed at examining the effects of cerebellar deep brain stimulation in a mouse model of ataxia. As the authors point out, ataxia and, in particular cerebellar ataxia, remains one of the disabling movement disorders without an adequate solution. In fact, one of the current partial exclusion criteria for offering patients deep brain stimulation for the management of movement disorders is the presence of ataxia. For example, when evaluating patients as potential candidates for the treatment of essential tremor, we tend to exclude those that presented with overlapping ataxia. Hence, this line of investigation is important not only to examine the opportunity to better characterize cerebellar pathways, but also to open new paths for a translational therapy that can be relevant to a patient population that today is under treated. The results of these experiments, including the main preclinical finding that corroborates the use of deep brain stimulation of the cerebellar nuclei for ataxia in the mouse model, are all very interesting. The putative mechanisms of action of cerebellar deep brain stimulation were thoroughly dissected and correlations were traced with several variables including age and viability of the Purkinje cell system. This reviewer has a few thoughts and concerns to be considered by the authors.

We are pleased that the reviewer found our manuscript to be detailed and clinically-relevant. We have incorporated the suggestions below in the revision of our manuscript.

Major Point #1: In the introduction and in the discussion the authors state that to date there are no paradigms that provide sustained benefits from deep brain stimulation after it is deactivated. This is not consistent with the practice and research of deep brain stimulation in movement disorders. While the carryover in patients with essential tremor can be minimal once electrical stimulation is stopped, carryover is present in patients with Parkinson's disease and dystonia. In fact, most studies evaluating the mechanisms underlying the effects of deep brain stimulation include washout periods in order to minimize the effects of carryover between phases of the research. This consideration is particularly significant in the management of dystonia. There is ample experience showing that dystonia does not immediately worsen once stimulation is discontinued as evidenced by patients that continue to do well for long periods of time after the pulse generator device dies. We suggest reconsidering this point around the novelty of carryover effects in deep brain stimulation.

Thank you for bringing this important point to our attention. We have revised our manuscript, namely our abstract and introduction, to take these suggestions into account. Accordingly, we have added appropriate references to highlight these studies in human disease. Based on such findings of the carryover effects in patients, we revised our argument to emphasize the importance of DBS targeting for consistent symptomatic relief over the lack of dedicated stimulation paradigms that elicit long-lasting improvements, and the need to mechanistically understand how the motor system might accommodate such changes.

Major Point #2: Still in the introduction, in the lines 84 and 85, the authors state that initial attempts for treating motor deficits with cerebellar DBS were inconsistent. Are the authors referring to the work by Ross and Cooper in the 1980s? If so, we note that those were neuromodulation studies but not deep brain stimulation studies as the investigators utilized surface electrodes on the tentorial surface of the cerebellum.

Excellent point. We have revised the sentence (page 4, lines 85-87) for clarification. It now reads, “While stimulating the cerebellar cortex was inconsistent for treating motor deficits¹⁹, stimulating the cerebellar nuclei has been more efficacious (EDEN; Improvement of Upper Extremity Hemiparesis Due to Ischemic Stroke: A Safety and Feasibility Study, ClinicalTrials.gov Identifier: NCT02835443)^{7,18}.”

Major Point #3: Deep brain stimulation target. The authors elected to stimulate the interposed nucleus of the cerebellum, which makes perfect sense given the spinocerebellar mechanisms related to ataxia. However, it may be difficult to target the interposed nucleus in the mouse and to restrict the effects of stimulation to such small nuclei. One option would be to frame this simply as deep cerebellar stimulation, acknowledging that the effects may be resulting from stimulation of the medial and lateral nuclei, combined. Another option would be to estimate the extent of the current spread relative to lead location, utilizing charge density and electrical stimulation modeling.

Agreed. In our revised manuscript, we replaced statements using the terminology “interposed DBS” with “cerebellar DBS” and in our discussion section we have also acknowledged that stimulation may extend to the fastigial and dentate nuclei (page 18, lines 409-411). Also, for a more complete description of how we have made these changes, please see our response to Major Point #5 from Reviewer #1, who provided us with a similar comment.

Major Point #4: The extensive examination of the effects of various frequencies of stimulation on behavior is probably the most impactful and interesting part of the study. The authors demonstrate the frequency-specificity of the results, which further provides internal controls to validate the most robust findings associated with low-frequency stimulation. One finding that stood out but is not followed by significant commentary is regarding the effects of 130 Hz stimulation on control mice. It seems that while low frequency stimulation improved motor control in ataxic animals, high-frequency stimulation enhanced motor control in normal controls. Is that the case? Please comment further on this very curious finding.

We are honored that you find our inclusion of different frequencies impactful. Indeed, we found that in certain DBS conditions control mice do improve. The control groups that improve (e.g., 0, 2, and 130 Hz with the addition of animals) started off significantly worse, likely due to a negative impact of the surgery. In such cases, DBS improves motor behavior, but the effects do plateau. We have included these details in an expanded discussion. Please also refer to our response to Reviewer #1’s Major Point #3, where we have also addressed a similar set of comments.

Major Point #5: Another very relevant finding is that DBS normalizes muscle activity in the ataxia model during locomotion. The authors demonstrate the effects of DBS on overlapping co-contractions of the tibialis anterior and gastrocnemius. We note that this type of agonist/antagonist co-contraction is very characteristic of dystonia and question if the CAR8 mouse model is really a pure model of ataxia or if it could be a model of ataxia and dystonia. This question is relevant to

the interpretation of the results as the authors may have found not only a therapeutic approach for ataxia but also a therapeutic approach that may be viable for some forms of dystonia.

You are absolutely correct. *Car8^{w^{dl}}* mice also have a subtle appendicular dystonia and a strong tremor, as we previously reported (White et al., 2016). For simplicity, our initial submission focused on the ataxia phenotype, but we now acknowledge that our EMG results could also be applicable to dystonia (page 22, lines 495-499). This is especially important given that our recent studies showed that stimulating the cerebellum (again, we targeted our electrodes to the interposed nuclei) resulted in improved motor function in mouse models of dystonia and tremor (White and Sillitoe, 2017; Brown et al., 2020).

Major Point #6: The authors investigated the effects of DBS according to age and provided data that correlate younger age with stronger therapeutic response. A co-variable is duration of disease, which can also influence response, as noted by the authors. While it makes perfect sense that animals with younger age or less duration of disease may have better outcomes, this reviewer is concerned with this small sample size that led to this conclusion. Based on figure 5, it seems that there are only one to three animals in each age group. We could not find statistical analysis supporting these assessments and there is a chance that the finding is really not significant. It's probably wiser to defer this topic to a future study that will be dedicated at examining the effects of age and disease duration on motor outcomes with low frequency deep brain stimulation in the mouse model of ataxia.

We extensively revised Fig. 5 to address this comment, as well as Minor Point #3 from Reviewer #1. This included stimulating more animals to increase the *n* of P60-P120 *Car8^{w^{dl}}* mice. Additionally, we made the following revisions: (1) We eliminated the timeline and replaced it with before-after bar graphs of P30 and \geq P150 mutants. (2) We revised our justification for using P30 and \geq P150 mice. We realize that what we originally wrote was not entirely reflective of why we stimulated young and old mutant mice. We primarily stimulated P30 and \geq P150 mice in order to determine whether the degree of motor function retained prior to stimulation influences outcomes, provided that the *Car8^{w^{dl}}* mice that started off performing better on the rotarod responded more robustly to beta-frequency DBS (Fig. 5, S14). Therefore, you will now see that we revised our language (page 14, paragraph starting at line 311) to more accurately reflect this. (3) We agree that investigating the specific effects of age and disease duration on motor outcomes would need to be studied more in-depth as a part of a future study and so we added sentences to address this in our discussion (pages 21-22, lines 483-486). We did not eliminate using the P30 and \geq P150 data from our manuscript because these results support our conclusion that residual motor function affects inter-subject variability and provides a foundation for our last figure, which explores potential explanations for the critical therapeutic window that we found.

Major Point #7: Lines 300 through 316. The authors tested the effects of interposed nucleus DBS in mutant mice that lack GABA neurotransmission, learning that there was no benefit of DBS in that model. Is this finding sufficient to conclude that DBS needs to be initiated before the peak symptoms of ataxia? Can the findings from this second mouse model directly inform the mechanisms of DBS in the CAR8 mutant mice? Perhaps a safer way to examine the role of GABA on the mechanisms of DBS in the CAR8 model would be to selectively inhibit GABA in this same

mouse model and evaluate if it alters the outcome. This reviewer is concerned about utilizing two different mouse models to infer the mechanisms of DBS in the first mouse model.

Thank you for these comments. We made several revisions to our manuscript to accommodate these points. First, our conclusion that DBS should be initiated prior to the peak of ataxia should have followed our Fig. 5 data and not our Fig. 6 data, where the *L7^{Cre};Vgat^{flox/flox}* mice were stimulated. Therefore, you will find in our revised manuscript that this conclusion (page 14, lines 323-325) is now following the Fig. 5 data. Second, we agree that we should be more careful on how we present *L7^{Cre};Vgat^{flox/flox}* data so that we are not inferring the mechanisms of DBS in *Car8^{wdl}* mice from data collected using another mouse model. To address this concern, we moved the *L7^{Cre};Vgat^{flox/flox}* data into its own results section, titled “Cerebellar DBS relies on Purkinje cell neurotransmission.” Please note that data from Fig. 5 and part of Fig. 6 are now discussed under the subheadings, “Ataxia severity contributes to the variability of DBS outcomes” and “Purkinje cell firing properties change with motor deterioration in *Car8^{wdl}* mice,” respectively. We also revised our justification and interpretation of the *L7^{Cre};Vgat^{flox/flox}* data in light of what has been previously published. For instance, in the new results section, we comment on how *Car8^{wdl}* symptomology correlates to alterations in Purkinje cell physiology (page 15, paragraph starting on line 328), then discuss how the most effective therapies to-date for ataxia regularizes cerebellar function (page 16, lines 363-365). This holds true across ataxia models (Jayabal et al., 2016; White et al., 2016; Alviña and Khodakhah, 2010). Therefore, we asked whether modulating Purkinje cell neurotransmission is necessary for treating ataxia. Our *L7^{Cre};Vgat^{flox/flox}* data suggest that DBS, like current investigational drugs for ataxia (Jayabal et al., 2016; White et al., 2016), depend on Purkinje neurotransmission. Finally, we found in a recent study that administering a tremorgenic drug to *L7^{Cre};Vgat^{flox/flox}* mice resulted in no behavioral change, showing that Purkinje cell neurotransmission is important for other behavioral adaptations as well (Brown et al., 2020). Therefore, we used the *L7^{Cre};Vgat^{flox/flox}* mice as a genetic method of manipulating the circuit. Still, as we also responded to Reviewer #1’s Major Point #11, we have noted that GABA-related molecular mechanisms may indeed be affected since GABA uptake into pre-synaptic vesicles is blocked in *L7^{Cre};Vgat^{flox/flox}* Purkinje cells (although GABA itself is not eliminated). We have been very deliberate to not make any conclusions about GABA function per se and how it might affect DBS. We conclude from this genetic mouse model that a closed-loop cerebellar circuit is needed to propagate or rescue motor dysfunction in ataxic mice that receive DBS.

REVIEWERS' COMMENTS

Reviewer #1 (Remarks to the Author):

General comments: The authors did a commendable job at addressing my major and minor critiques. I only have two minor considerations that can help to fill in small gaps in the manuscript.

Minor point#1: Ideally, figure 1f should have an asterisk or another symbol to denote significance at 13 Hz to be consistent with figure 1e.

Minor point#2: Page 9, line 196: "We found that cerebellar stimulation increases slow-twitch fiber protein expression". Yet, the p values show otherwise ($p=0.1011$; $p=0.1252$). Also, figure S9 includes pie-charts related to this statement. However, the statistical analyses performed is not described in the figure caption or manuscript. Please clarify.

Reviewer #2 (Remarks to the Author):

I support publication of this manuscript. The authors did a nice job with addressing reviewer concerns.

Reviewer #3 (Remarks to the Author):

Thank you for the revisions. I have no other comments

Response to reviewer comments for manuscript NCOMMS-20-12228B now entitled "**Neuromodulation of the cerebellum rescues movement in a mouse model of ataxia**" performed by Lauren N. Miterko et al.

We would like to thank the reviewers for their final comments.

Below are our explanations for how we have altered the manuscript in this revised version. The Reviewer's comments are written in regular text, and our responses are written in bold.

Reviewer #1: The authors did a commendable job at addressing my major and minor critiques. I only have two minor considerations that can help to fill in small gaps in the manuscript.

Thank you. We appreciate the suggestions that Reviewer #1 had made that helped us strengthen our manuscript. We addressed the remaining concerns below.

Minor Point #1: Ideally, figure 1f should have an asterisk or another symbol to denote significance at 13 Hz to be consistent with Figure 1e.

We have added an asterisk to Fig. 1f to denote significance at 13 Hz.

Minor Point #2: Page 9, line 196: "We found that cerebellar stimulation increases slow-twitch fiber protein expression." Yet, the p values show otherwise (p=0.1011; p=0.1252). Also, Figure S9 includes pie charts related to this statement. However, the statistical analyses performed is not described in the figure caption or manuscript. Please clarify.

Thank you for catching this, we have updated our text. We have included the statistical test we performed in the Methods (pgs. 40-41, lines 915-919, under "Image Acquisition and Quantification." We performed a Three-Way ANOVA using genotype, definitive fiber type (fast vs. slow), and stimulation frequency (0 vs. 13 Hz) as the independent variables and found that slow-twitch fiber protein expression is significantly increased in 13 Hz-stimulated *Car8^{w^{dl}}* mice (p=0.0312).

We added the Three-Way ANOVA and the post-hoc test performed (Sidak's multiple comparisons test) to the Figure Legend (Supplemental Information File).

Reviewer #2: I support the publication of this manuscript. The authors did a nice job with addressing reviewer concerns.

We are pleased that the reviewer supports the publication of our revised manuscript.

Reviewer #3: Thank you for your revisions. I have no other comments.

We are pleased that the reviewer found our revisions to be satisfactory.